



# Cataclastic deformation of triaxially deformed, cemented mudrock (Callovo Oxfordian Clay): an experimental study at the micro/nano scale using BIB-SEM

Guillaume Desbois[1], Nadine Höhne[1], Janos L. Urai[1], Pierre Bésuelle[2], Gioacchino Viggiani[3]

[1]*Structural Geology, Tectonics and Geomechanics, RWTH Aachen University, Lochnerstrasse 4-20, 52056 Aachen, Germany*

[2]*CNRS, 3SR, Grenoble, France*

[3]*Université Grenoble Alpes, 3SR, Grenoble, France*

## Abstract

The macroscopic description of deformation and fluid flow in mudrocks can be improved by a better understanding of microphysical deformation mechanisms. Here we use a combination of Scanning Electron Microscopy (SEM) and Broad Ion Beam (BIB) polishing to study the evolution of micro structure in samples of Callovo-Oxfordian Clay that were previously tested in the lab. Digital Image Correlation (DIC) enabled for the measurement of strain fields in the specimens, which were used as a guide to select regions in the sample for BIB-SEM analysis. Microstructures show evidence for dominantly cataclastic mechanisms (intergranular, transgranular, intragranular cracking, grain rotation, clay particle bending) down to nm- scale. At low strain, the dilatant fabric contains individually recognizable open fractures, while at high strain the reworked clay gouge contains broken non-clay grains, with a clear change towards smaller pores than the undeformed material and corresponding resealing of initial fracture porosity.

This study might provide a first step towards a micro scale basis for constitutive models of deformation and fluid flow in cemented mudstones

**Keywords:** cemented mudrock, Callovo Oxfordian Clay, triaxial deformation, clay microstructure, deformation mechanisms, BIB-SEM, DIC, cataclastic deformation

## 1 Introduction

Mudrocks constitute up to 80% of the Earth's sedimentary rocks (Stow, 1981). Due to their low permeability and self-sealing properties (Boisson, 2005, Bernier et al., 2007), claystones are





considered for nuclear waste disposal and seals for storage in deep geological formations (Salters &
Verhoef, 1980; Shapira 1989; Neerdael & Booyazis, 1997; Bonin, 1998; ONDRAF/NIRAS, 2001;
NAGRA, 2002; NEA, 2004; ANDRA 2005; IAEA, 2008, Ingram & Urai, 1999). Studies on
mechanical and transport properties over long time scales are essential to the evaluation of subsurface
integrity. For this, it is generally agreed that a multiscale experimental approach that combines the
measurement of the bulk mechanical and transport properties of the specimen with microstructural
study to identify deformation mechanisms is required to develop microphysics-based constitutive
equations, which can be extrapolated to time scales not available in the laboratory, after comparison
with naturally deformed specimens (Morgenstern & Tchalenko 1967; Tchalenko, 1968; Lupini et al.,
1981; Rutter et al., 1986; Logan et al., 1979, 1987, 1992; Marone & Scholz, 1989; Evans & Wong,
1992; ; Katz & Reches, 2004; Colletini et al., 2009; Haines et al., 2009, 2013; French et al., 2015;
Crider, 2015; Ishi, 2016).
In the field of rock mechanics and rock engineering, experiments are performed to low strain and short
term in order to predict damage and deformation in tunnelling and mining, for example. Here, a
macroscopic and phenomenological approach is common, to characterize mechanical and transport
properties and to establish the constitutive laws. Microstructures are rarely studied because the
strained regions are difficult to find (except macroscopic fractures), and because microstructures
below micrometre scales are elusive. However, a microphysics-based understanding of mechanical
and fluid flow properties in mudrocks provides a better basis for extrapolating constitutive equations
beyond the time scales accessible in the laboratory, so that predictions over the long term can be made
less uncertain. This requires integration of measurement of the mechanical and transport properties
with microstructures, towards multi-scale description of deformation in mudrocks at low strain.
The microstructural geology community studied microstructures in deformed mudrocks to infer
deformation mechanisms (Dehandschutter et al., 2004; Gratier et al., 2004; Klinkenberg et al; 2009;
Renard, 2012; Robinet et al., 2012; Richard et al., 2015; Kaufhold et al., 2016), but this was limited by
the problems with sample preparation. The mechanical properties and related microstructures of
natural and experimental high strain fault rocks have been studied extensively (Bos & Spiers, 2001;
Faulkner et al., 2003, Marone & Scholz, 1989). For Opalinus Clay (OPA) deformed in laboratory,
Nüesch (1991) and Jordan and Nüesch (1989) concluded that cataclastic flow was the main
deformation mechanism, with kinking and shearing on R- and P-surfaces at the micro scale, however
this was only based on observations with optical microscopy, so the grain scale processes were not
resolved. Klinkerberg et al. (2009), demonstrated a correlation between compressive strength and
carbonate content of two claystones this correlation is positive for OPA but negative for Callovo
Oxfordian Clays (COX). This was explained by the differences in grain size, shape, and spatial
distribution of the carbonate (Klinkerberg et al. 2009), cf. Bauer-Plaindoux et al. (1998).



Microstructural investigations using BIB-SEM and FIB-TEM in OPA from the main fault in the Mt-
Terri Underground Research Laboratory (Laurich et al., 2014) showed that inter- and transgranular
microcracking, pressure solution, clay neoformation, crystal plasticity and grain boundary sliding are
playing an important role in micro-scale processes during the early stages of faulting in OPA.
Cataclastic microstructures are rare and there was an almost complete loss of porosity in micro- shear
zones.
Digital Image Correlation (DIC) during experimental deformation, in 2D or 3D, measured the
displacement fields and quantifies strain over time (Lenoir et al., 2007 Bornert et al., 2010; Bésuelle &
Hall, 2011; Dautriat et al., 2011; Wang et al., 2013, 2015; Fauchille et al., 2015; Sone et al., 2015). In
samples of rock salt, sand or mudrocks, processes occurring at grain scale can now be studied with
high resolution (Hall et al., 2010; Andò et al., 2012; Bourcier et al., 2012, 2013; Wang et al., 2015).
On claystones, DIC was used to study swelling in environmental SEM (Wang et al., 2013, 2015;
Fauchille et al., 2015) to measure strain at between clay matrix and non clay minerals.
Microstructural studies in mudrocks are currently in rapid development driven by the development of
ion beam milling tools (e.g. Focussed Gallium Ion Beam (FIB) and Broad Argon Ion Beam (BIB)
which allow imaging of mineral fabrics and porosity down to nm- scale in very high quality cross
sections with SEM and TEM (Lee et al., 2003; Desbois et al., 2009, 2011, 2013, 2016; Loucks et al.,
2009; Curtis et al., 2010; Heath et al., 2011; Klaver et al., 2012; Keller et al., 2011, 2013; Houben et
al., 2013, 2014; Hemes et al., 2013, 2015; Laurich et al., 2014; Warr et al., 2014; Song et al., 2016).
Serial sectioning allows reconstruction of microstructure in 3D, and cryogenic techniques can image
the pore fluid in the samples (Desbois et al., 2013, 2014; Schmatz et al., 2015).
Previous work has shown that the mechanical properties of Callovo Oxfordian Clays (COX) do not
only depend on the fraction and mineralogy of the clay but also on water content and texture (Bauer-
Plaindoux et al., 1998). Chiarelli et al. (2000) showed that COX is more brittle with increasing calcite
content and more ductile with increasing clay content and proposed two deformation mechanisms:
plasticity induced by slip of clay sheets and induced anisotropic damage as indicated by microcracks
at the interface between grains and matrix, however they provided very little microstructural evidence
to support this. Gasc-Barbier et al., (2004), Fabre et al., (2006), Chiarelli et al., (2003), Fouché et al.,
(2004) report that the COX has an unconfined compressive strength of 20 to 30 MPa and a Young's
modulus of 2 to 5 GPa. In the context of underground storage of radioactive wastes, these papers try to
predict the mechanical evolution of COX over the period of thousands of years. The effects studied
include creep, pore-pressure dissipation, swelling, contraction, chemical effects, pressure solution and
force of crystallization. Although these papers develop elaborate constitutive laws, they provide very
limited microstructural observations. The need for micromechanical observations was already
recognized by Yang et al. (2012) and Wang et al., (2013, 2015) who have conducted deformation





experiments under ESEM observation combined with Digital Image Correlation. They have shown
how heterogeneous strain fields correlate with microstructure and recognized shear bands and tensile
microcracks.
For highly overconsolidated claystones from the Variscan foreland thrust belt in the Ardennes and
Eifel, Holland et al. (2006) proposed an evolutionary model of the fault zone that developed mainly by
mechanical fragmentation of the original claystone's fabric with only minor contribution by diagenetic
changes or weathering. In this model, the initial loss of cohesion is driven by kinking, folding and
micro-fracturing processes related to an increasing porosity and possibly permeability. Abrasion
during progressive deformation increases the amount of clay gouge material, and re-sealing by
decrease in pore size and porosity in the clay gouge.
In summary, deformation mechanisms in mudrocks are poorly understood. Although as a first
approximation the plasticity of cemented and uncemented mudrocks can be described by somewhat
similar, pressure dependent constitutive models, the full description of their complex deformation and
transport properties would be much improved by better understanding of the microscale deformation
mechanisms. There is a wide range of possible mechanisms:  intra- and intergranular fracturing,
cataclasis, grain boundary sliding, grain rotation and granular flow, crystal plasticity of clays, a the
poorly known plasticity of nano-clay aggregates with strong role of clay-bound water, and
cementation, fracture sealing and solution- precipitation.
This contribution combines stress-strain data, measurement of displacement fields by digital image
correlation (DIC) with microstructural investigations in selected areas based on the DIC results. For
this, we prepared square millimetre-sized high quality cross sections by broad-ion-beam milling (BIB)
followed by scanning electron microscopy (SEM) this allows us to infer microphysical processes of
deformation with sub-micron resolution (Figure 1). The two samples used are from the Callovo-
Oxfordian Clay (COX, a cemented claystone):  one deformed in plane strain compression at 2 MPa
confining pressure (COX-2MPa, (Bésuelle & Hall, 2011) and another in triaxial compression at 10
MPa confining pressure (COX-10MPa, Lenoir et al., 2007).  Specimens were taken from the Bure site
in Meuse-Haute Marne in France, and belong to the clay-rich facies of COX.

## 2    Material studied and DIC-derived strain fields

Mechanical experiments were performed on two COX samples collected at the ANDRA Underground
Research Laboratory located at Bure (Meuse/ Haute Marne, Eastern France) at approximately 550 m
below the ground surface (Boisson, 2005). The clay fraction (illite/smectite, illite, chlorite) is 40–45%,
carbonate (mostly calcite) and quartz 25–35% and 30%, respectively and minor feldspar, mica and
pyrite.



The details of these experiments including instrumentation set-ups, boundary conditions and DIC
interpretations are comprehensively described in Bésuelle and Hall (2011) and Lenoir et al. (2007).
This contribution presents mostly the microstructural analysis performed on these previously
deformed two samples.
The first sample considered in this study (COX-2MPa) was tested in plane strain compression at 2
MPa confining pressure. 2D DIC was performed on consecutive photographs of one side of the
specimen (in the plane of deformation) throughout the test. Further details are given in Bésuelle and
Hall (2011). The second sample (COX-10MPa) was tested in triaxial compression at 10 MPa
confining pressure. 3D DIC was performed on consecutive x-ray images of the specimen obtained in a
synchrotron throughout the test. Further details are given in Lenoir et al. (2007) – please note that in
this publication this sample is referred to as ESTSYN01.
In the following, the relevant findings in Bésuelle & Hall (2011) and Lenoir et al. (2007) are
summarized:
(1) The prismatic sample COX-2MPa was tested in plane strain compression in a true triaxial

apparatus at a constant value of $\sigma_3 = 2$ MPa. The size of the specimen is 50 mm in the vertical

direction, which is the direction of major principal stress (1), 30 mm in the direction of

intermediate principal stress (2), and 25 mm in the direction of minor principal stress (3). The

test was displacement-controlled, with a constant rate of displacement (in the direction 1) of

1.25 μm/s, i.e., a strain rate of $2.5 \ 10^{-5} \ s^{-1}$ (see Bésuelle & Hall 2011 for further details). Figure

2a shows the evolution of the differential stress ($\sigma_1 - \sigma_3$) vs. axial strain (specimen shortening

divided by its initial height). The curve shows a first stress peak at 0.02 axial strain, followed

by a strong stress drop. Then, a slow stress increase is observed, followed by a second stress

drop at 0.42 axial strain. After, the stress is quite constant. As shown in Figures 2b and 2c,

these two stress drops are associated with major failure by faulting in the specimen. The crack

that appeared during the second drop is conjugate to the first crack set, which appeared at the

first drop. This set of conjugate fractures, at an angle of 20° to 45° about the direction 1, will

be referred to as "main synthetic fractures" in the following sections. The DIC-derived strain

fields in Figures 2b and 2c also show that the development of each single conjugate fracture is

accompanied by relay zones with a set of antithetic fractures. Moreover, the fracture appearing

during the second stress drop (Fig. 2c) is also reactivating the first fracture and its associated

antithetic fractures. At this resolution (pixel size is 10 μm), the set of conjugate fractures and

the associated antithetic fractures are the major features of localized deformation: they

represent zones where the sample was sheared with damaged zones having a thickness of





about 60 μm. Dilatancy was also measured in the damaged zones mentioned above (see
volumetric strain fields, Figs. 2b and 2c).

(2) The cylindrical sample COX-10MPa (10 mm in diameter and 20 mm in height) was sheared in
triaxial compression at a confining pressure of 10 MPa. The test was carried out under
tomographic monitoring at the European Synchrotron Radiation Facility (ESRF) in Grenoble,
(France), using an original experimental set-up developed at Laboratoire 3SR at the University
of Grenoble Alpes (France).. Complete 3D images of the specimens were recorded throughout
the test using x-ray microtomography (voxel size was 14 μm). The test was displacement-
controlled, with a displacement rate of 0.05 μm/s, i.e., an axial strain rate of $2.5 \times 10^{-6}$ s$^{-1}$. The
stress-strain curve (Figure 3.a) shows only one stress peak at an axial strain of 0.04. The peak
stress is followed by a major stress drop corresponding to the formation of a shear fracture
(referred to as "main synthetic fracture" in the following sections) oriented at an angle of 30-
40° about the direction of the principal stress $\sigma_1$ (the DIC-based maximum shear strain fields
are given in Fig. 3b). The DIC-derived volumetric strain fields (not shown here, see Lenoir
2006) indicate that the shear fracture is accompanied by some slight dilatancy.

## 3    Methods: BIB-SEM imaging of deformed microstructures

Sub-samples were selected to represent areas with different strain history based on the DIC analysis.
For COX-2MPa, three BIB cross sections were prepared around the conjugate fractures in areas with
different amount of diffuse strain (at the resolution of DIC), antithetic fractures (ROI-2, ROI-3 and
ROI-4; Figures 4.b, 6.b, c, d and 7) and a fourth one in a region without measurable strain (ROI-1;
Figures 4.b and 6.a). For COX-10MPa, two BIB-SEM analyses were done around the single shear
fracture (Figures 4.e and 6.e, f).
Sub-samples were first stabilized with epoxy, extracted with a low speed diamond saw in dry
conditions, pre-polished dry using SiC polishing papers and BIB polished by using a JEOL SM-09010
cross-section polisher (for 8 h, $1.10^{-3} – 1.10^{-4}$ Pa, 6 kV, 150 μA). BIB cross-sections are all prepared
parallel both to the principle stress ($\sigma_1$) direction and perpendicular to the shear displacement plane.
BIB cross sections of about 1.5 mm$^2$ (Figures 6 and 7) were imaged with a Supra 55-Zeiss SEM (SE2
and BSE detectors at 20 kV and WD = 8 mm). Further details of the method are given in (Klaver et
al., 2012, 2015, Houben et al., 2013, 2014, Hemes et al., 2013, 2015, Desbois et al., 2016)




## 4   Results

### 4.1   Overview of microstructures

The sub-sample without measurable strain (i.e. ROI-1_COX-2MPa, Figure 6a) shows non-clay minerals in a clay matrix with a weak shape preferred orientation parallel to bedding (perpendicular to the experimental $\sigma_1$). The clay matrix contains submicron pores typical of compaction and diagenesis, with a power law distribution of pore sizes. Pores commonly have very high aspect ratio, with the long axis oriented sub-parallel to the bedding. These characteristics are very similar to those in the undeformed COX sample (Figure 5, cf. Robinet et al. 2012).

In all other BIB cross-sections (Figures 6.c-f and 7), both samples show damaged microstructures. At the sample scale, three different types of fracture are identified: (i) the main synthetic fracture (section 2), (ii) antithetic fractures (Figure 6) and (iii) joints sub-parallel to the main fracture. The material between the fractures zones has very similar microstructure to the undeformed COX.

### 4.2   Detailed description of microstructures

#### 4.2.1   *Arrays of antithetic fractures*

In COX-2MPa the antithetic fractures (Figure 7) are of two different types. *Type I* is located only in the clay matrix (Figure 8.a), with apertures up to few micrometres, with boundaries closely matching - suggesting that these are opening mode fractures (Mode I). *Type II* fractures consist of a damage zone with thickness up to 25 µm (Figure 8.e, f, g, h, i) containing angular fragments of non-clay minerals and clay aggregates, (Figure 8.h), sometimes with preferred orientation parallel to the fracture. The transition between the damage zone and the undeformed host rock is sharp (Figure 8.f, g, h, i). In relay zones the fracturing becomes so intense that the clay matrix is fragmented into clay-size fragments (Figure 8.i). Porosity in these relay zones is locally much higher and pores much larger than in undeformed COX. Fracture boundaries usually do not match (Figure 8.h). Figure 8.e shows examples where parts of broken non-clay minerals can be matched.

COX-10MPa, we observed the two types of antithetic fractures mentioned above. Antithetic fractures of *Type I* are very similar (indicated in Figure 6.f) to those in COX-2MPa but rare, whereas antithetic fractures of *Type II* contain a wider damage zone in comparison to those in COX-2MPa, in which the average grain size and the pore size is significantly smaller, consistent with stronger cataclasis at high confining pressure. In parts of the damage zones interpreted to be restraining sections, pores in the reworked clay aggregates cannot be resolved in the SEM.



In both samples, the fragments between the arrays of antithetic fractures show only minor deformation
indicated by fractured grains of organic matter (Figure 8.b), calcite (Figure 8.d, c) or quartz (Figure
8.d). Visible relative rotation of parts of fractured grain is rare (Figure 8.d).

### 4.2.2    Synthetic fractures

The synthetic fractures are the regions that localized most of strain and have the thickest damage zone
(Figures 2 and 3). Here, COX-2MPa and COX-10MPa show very similar microstructures. The grain
(fragment) size of non-clay minerals is significantly smaller than in the host rock and their sizes are
poorly sorted. Non-clay minerals have also angular or/and chipped edges (Figures 9, 10 and 12).
Locally, grains in the damaged zone show trans-granular fractures (Figure 10.c and 12.a). In parts of
the damage zone, dilatancy and a strong increase in connected porosity (ROI-4_COX-2MPa, Figure 9
and ROI-2_COX-10MPa, Figure 10) is indicated by epoxy impregnation.    In other parts, (ROI-
1_COX-10MPa) strongly reworked clay matrix is not impregnated and shows no pores visible at the
resolution of image (83.8 nm pixel size in Figure 11.b, c).
For COX-2MPa, the DIC analysis shows that the conjugated synthetic fractures form a complex
network of fracture's branches at region where they both intersect (Figure 2.c). The ROI-3_COX-
2MPa sub-sample (Figure 4.c) covers two of these branches. Microstructural analysis of these two
branches in ROI-3_COX-2MPa show similar microstructures, with only the fracture apertures being
different (Figure 6.c).
In both COX-2MPa and COX-10MPa, the damage zone of the synthetic fractures contains an open
fracture (Figures 9, 10 and 12), with apertures of 50 - 70 µm. These large open fractures are filled with
epoxy, have matching boundaries and never crosscut the non-clay minerals in the damage zone.
Similar fractures are found in COX-2MPa but parallel to the antithetic fractures, with jagged
morphologies, matching walls never crossing the non-clay minerals (Figure 8.b, c, e). These fractures
are not resolved by DIC.

## 5    Discussion

### 5.1    Artefacts caused by drying and unloading

Claystones are sensitive to changes in hydric conditions that can lead to the shrinkage or the swelling
of the clay matrix (Galle, 2001; Kang et al., 2003; Soe et al., 2007; Gasc-Barbier & Tessier, 2007;
Cosenza et al., 2007; Pineda et al., 2010; Hedan et al., 2012; Renard, 2012; Desbois et al., 2014).





The DIC analysis is not affected by this because the images were acquired during deformation of
preserved (wet) samples. SEM analysis is done on samples which have been deformed and unloaded,
followed by slow drying and further dehydration in the high vacuum of the BIB and SEM. In COX-
10MPa, this is illustrated by Figures 4.d and 4.e. Figure 4.d shows the sample at the end of the
deformation experiment, whereas Figure 4.e shows the same sample but about 10 years later, both
imaged with X-ray. The comparison of Figures 4.e and 4.e shows that cracks developed parallel to the
bedding and that the apertures of fractures developed during the deformation became larger. These are
interpreted to result from unloading and shrinkage during drying and aging of specimens. We infer
that similar changes occurred in COX-2MPa: the wider damage zones in conjugate synthetic fractures
imaged by SEM (Figure 6.c, d) compared the width estimated from the DIC analysis corroborates this
interpretation.
The fractures in the fragments between the arrays of antithetic fractures (Figure 8.b,c,d) are not
present in the low strain ROI-1_COX-2MPa, and they are sub parallel to $\sigma_1$, strongly suggesting they
are formed by experimental deformation.
In contrast, antithetic fractures of *Type II* (Figures 6, 7 and 8.e-i) are interpreted to develop during
deformation because: (i) the internal microstructures and fabrics are damaged and (ii) DIC recorded a
clear localization of strain in these. Though the antithetic fractures of *Type I* are not clearly recognized
at the resolution of DIC, most of these in COX-2MPa (Figure 8.a) are interpreted to develop during
deformation because they are oblique to the bedding and parallel to the antithetic fractures of *Type II*
(Figure 6, 7 and 8.f-g). One exception is the antithetic fractures of *Type I* observed in ROI-1_COX-
10MPa (Figure 6.e), which is parallel to bedding. Mode I fractures sub-parallel to the main synthetic
fractures are less easy to interpret: they may be related to the rotation of blocks between the antithetic
fractures (Kim et al., 2004). Cryogenic techniques to preserve wet fabrics combined with ion beam
milling and cryo-SEM (Desbois et al., 2008, 2009, 2013, 2014) is the dedicated technique to solve this
question in the future.
**5.2   Deformation mechanisms**
The first, and perhaps surprising conclusion based on the observations above is the dominantly
cataclastic deformation in Callovo-Oxfordian Clay at confining pressures up to 10 MPa.
Microfracturing, producing fragments at a range of scales and reworking into a clay-rich cataclastic
gouge during frictional flow are the main processes in both samples. This is accompanied by dilatancy
by microfracturing of the original fabric, but also by progressive decrease of porosity and pore size in
the gouge with the non-clay particles embedded in reworked clay.





Although in many cases the initial fractures propagate around the hard non-clay grains, there is also
significant fracturing of the hard non-clay minerals (e.g. Figure 8.b-d). This can be due to local stress
concentrations at contacts between adjacent non-clay clay minerals, or because the clay matrix is so
strongly cemented that it can transmit stresses sufficient to fracture calcite and quartz grains. Broken
non-clay minerals can displace or rotate with respect to each other (Figure 8.d) with local dilatancy
during deformation (Figure 2.b), in agreement with the interpretation of DIC measurements in
Bésuelle & Hall (2011) and Lenoir et al. (2007).
In COX-2MPa, the propagation of antithetic fractures of Type I (Figure 8.a) is predominantly in the
clay matrix (Figure 8.a). This is in agreement with the smaller strain in comparison to antithetic
fractures of Type II. Antithetic fractures of Type II contain angular non-clay grains with size smaller
than those in the host rock. We interpret these as evidence for comminution by grain fracturing.
Matching broken grains (Figure 8.e) are rare and in agreement with high strain cataclastic flow.
Fragments of clay aggregates in the antithetic fractures of Type II are much less coherent (Figure 8.h)
and more porous than the undeformed COX (Figure 8.i), indicating strong remolding by cataclastic
flow, and perhaps also plastic deformation of phyllosilicates. Here, pore morphologies are not
compatible with drying - induced deformation only, and we infer that these developed during
deformation.
Microstructures in the main synthetic fractures, both in COX-2MPa (Figure 9) and COX-10 MPa
(Figures 10 and 12), are similar. Angular non-clay minerals in the reworked clay matrix have a wide
range of grain sizes, smaller than those in the host rock. These characteristics are typical for cataclasis
(Passchier & Trouw, 2005). COX-2MPa the cataclastic gouge seems to be more porous than in COX-
10 MPa; this is as expected for the lower mean stress, but firm conclusion require further study to
exclude that this is an unloading and drying effect. For COX-10MPa, the porosity in clay matrix is
clearly reduced in comparison to the one in the host rock: most pores, if present, are below the
resolution of SEM (Figure 10 and 11). The mechanism of this compaction during shearing is
interpreted to be a combination of cataclasis of the cemented clay matrix, and shear-induced
rearranged the clay particles around the fragments of non-clay particles.

## 312   6   Conclusions

The integration of bulk stress-strain data, the analysis of displacement fields by 3D digital image
correlation (DIC) with Broad Ion Beam cutting and Scanning Electron Microscopy (BIB-SEM) is a
powerful multiscale method to study the deformation behaviour of mudstones.





We studied samples of Callovo-Oxfordian Clay (COX) subjected to triaxial compression at 2 MPa
and 10MPa confining pressure. DIC was used to locate regions deformed to different states of strain
and BIB-SEM allows microstructural investigations of mineral and porosity fabrics down to
nanometre scale.
Microstructures show evidence for dominantly cataclastic mechanisms (intergranular, transgranular,
intragranular cracking, grain rotation, clay particle bending) down to nm- scale.
At low strain, the dilatant fabric contains individually recognizable open fractures, while at high strain
in shear fractures the reworked clay gouge evolves towards smaller pores than the undeformed
material and corresponding resealing of initial fracture porosity. This shear induced resealing is more
important at the higher confining pressure.
This study provides a first step towards a microphysical basis for constitutive models of deformation
and fluid flow in cemented mudstones, with an improved extrapolation of these models for long time
scales.
In the future, the microstructures on experimentally deformed specimens needs to be compared with
the microstructures in naturally deformed claystones (Laurich et al.; 2014) in order to extrapolate the
constitutive models to long time scales.
**Acknowledgements**
We thank ANDRA for providing samples.

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




## Figure captions

**Figure 1:** Drawing of the experimental concept used for the investigation of experimentally deformed fine-grained mudrocks from bulk-scale to nm-scale. The example is based on a triaxial deformation test (10 MPa confining pressure) performed on a cylindrical Callovo-Oxfordian Clay, where displacement fields were followed by volumetric DIC on X-ray microtomography images (after Lenoir et al., 2007;).

**Figure 2:** Results of deformation test done on sample COX-2MPa. (a): deviator stress vs. axial strain response. The red star indicates the state of sample when BIB-SEM microstructural analyses are done. (b) and (c): incremental volumetric strain fields (VSF) and maximum shear strain fields (SSF) fields for deformation increment 1-2 and 3-4 indicated in (a) after DIC. After Bésuelle et al. (2011).

**Figure 3:** Results of deformation test done on sample COX-10MPa. (a): deviator stress vs. axial strain response. The red star indicates the state of sample when BIB-SEM microstructural analyses are done. (b) and (c): incremental maximum shear strain fields for deformation increment 1-2 and 2-3 indicated in (a) interpreted after DIC. After Lenoir et al. (2007).

**Figure 4:** Selection of differently strained areas (ROI) highlighted from DIC analysis on samples COX-2MPa (a-b) and COX-10MPa (c-e), for BIB-SEM microstructural analyses. (b): for COX-2MPa, four ROI were analysed: three at conjugate synthetic fractures in areas with different amount of diffuse strain and antithetic fractures (ROI-2, ROI-3 and ROI-4) and one in a region without measurable strain. (e): for COX-10MPa, two ROI were analysed both around the single synthetic shear fracture. (d) shows the X-ray radiography of the sample taken directly at the end of the deformation test. (e) shows the X-ray radiography of the same sample but taken about 10 years after the end of the deformation: drying cracks developed following the bedding and the aperture of the single shear fracture became larger. Orientation of $\sigma_1$ and of the bedding are indicated in red.

**Figure 5:** (a): BSE SEM micrograph of the typical mineral fabric in undeformed COX. (b): SE2 SEM micrograph of a detail from (a) showing the typical pore fabric in undeformed COX.

**Figure 6:** BSE SEM micrographs of the BIB cross-sections' overviews of COX-2MPa (a-d) and COX-10MPa (e-f) at differently strained areas (ROI) highlighted from DIC analysis in Figure 4. High strained ROI (c-f), display damaged microstructures, where three different types of fracture are identified: (1) the main synthetic fracture, (2) antithetic fractures oriented about 60º to the main fracture and (3) joints sub-parallel to the main synthetic fracture. These fracture are respectively indicated by 1, 2, 3 numbers in the figure. Orientation of the principle stress ($\sigma_1$) is indicated in red. Dashed yellow lines indicate the boundaries of the BIB polished areas.

**Figure 7:** Larger field of BSE SEM micrograph of the BIB cross-section's overview at ROI-1 in COX-2MPa sample. It shows the network of antithetic fractures (indicated by number 1) oblique to the principle main synthetic fracture (indicated by number 2). Orientation of the principle stress ($\sigma_1$) is indicated in red. Dashed yellow lines indicate the boundaries of the BIB polished areas.





**Figure 8:** Detailed microstructures in sample COX-2MPa. (a): a fracture running parallel to the antithetic fractures and at the interfaces between non-clay mineral and clay matrix. (b) and (c): intragranular fractures (i) and transgranular fractures (ii) at impingement of non-clay minerals. (d): a broken quartz grain showing evidence for rotation of its broken fragments. (e): incipient of flow of broken non-clay mineral within the antithetic fractures. (f) and (g): parts of antithetic fractures displaying thick damaged fabrics made of broken grains and clay matrix fragments. (h): Detail from (g). (i): Detail from (f) showing the denser and deformed fabric of a part of the clay matrix squeezed between a quartz grain located in the damaged fabric and the boundary with the host rock. In (f-i), the damaged zone is related to a higher porosity in comparison to the host rock. Orientation of $\sigma_1$ is indicated in red. Dashed yellow lines indicate the boundaries between the damaged fabric and the host rock, and also some grain boundaries. Qtz: quartz; Cc.: calcite; OM: organic matter. Black squares in (f) are missing pictures.

**Figure 9:** Detailed microstructure close the main fracture (indicated by number 1) in sample COX-2MPa. The main fracture displays internal damaged fabric made of fragments of broken non-clay minerals and clay matrix. Close to the main synthetic fracture, the host rock displays jagged joints sub-parallel to the main synthetic fracture (indicated by number 3) starting and ending at antithetic fracture (indicated by number 2). Orientation of the principle stress ($\sigma_1$) is indicated in red. The dashed yellow line indicates the boundary between the damaged fabric and the host rock.

**Figure 10:** Microstructures of ROI-1 in sample COX-10MPa. (a-c): The damaged fabric within the main fracture (1) is made of fragments of non-clay minerals derived from the dense, tight clay matrix. (a): the large open fracture in the middle of the main fracture (black) is interpreted to develop after the experiment by unloading or/and drying (see Section 5.1 for details). (c): some grains within the damaged fabric, but close to the boundary between the damaged fabric and the host rock, show transgranular fracturing (ii). Orientation of the principle stress ($\sigma_1$) is indicated in red. The dashed yellow lines indicate the boundaries between the damaged fabric and the host rock.

**Figure 11:** Detail of Figure 10.b. Microstructures (ROI-1_COX-10MPa) showing detail of porosity in BSE SEM micrograph (a) and SE2 SEM micrograph (b). At the resolution of the SEM micrograph, the damaged fabric appears very low porous in comparison to the host rock. The dashed yellow line indicates the boundary between the damaged fabric (left) and the host rock (right).

**Figure 12:** Detailed microstructures at ROI-2 in sample COX-10MPa. (a-c): The damaged fabric within the main synthetic fracture (indicated by number 1) is made of fragments of non-clay minerals and clay matrix derived from the host rock. (a): some grains within the damaged fabric, but close to the boundary between the damaged fabric and the host rock, show transgranular fracturing (ii). Detailed observations in (b) and (c) (SE2 SEM and BSE SEM micrographs of the same sub-area, respectively) show that parts of the damaged fabric display (1) porous island, where pores are between the fragments of non-clay and clay matrix; whereas other parts display (2) low porous islands made of fragment of non-clay minerals embedded in a dense, tight clay matrix. Pores within the porous island can be either filled with epoxy (in deep black pixel values) or not.





Orientation of $\sigma_1$ is indicated in red. The dashed yellow lines indicate the boundaries between the damaged
fabric and the host rock.
**Figures**






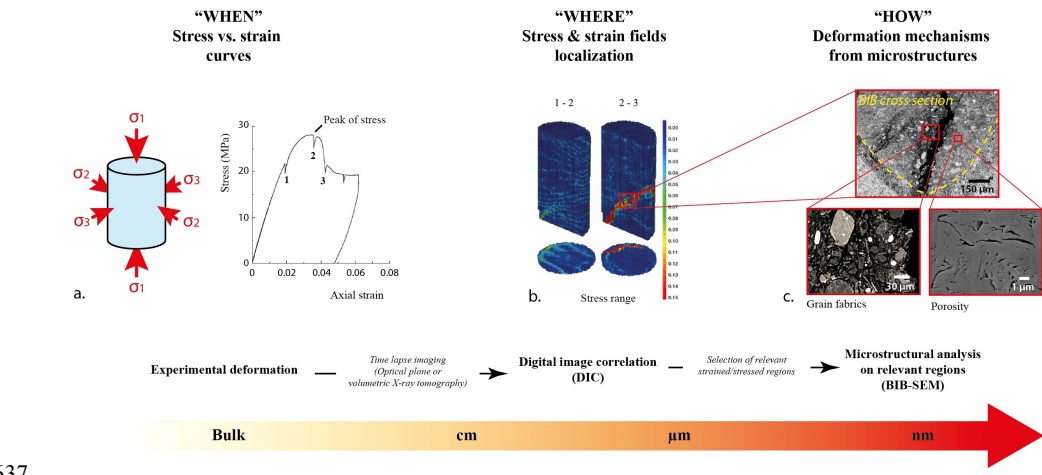


638                                          **Figure 1**













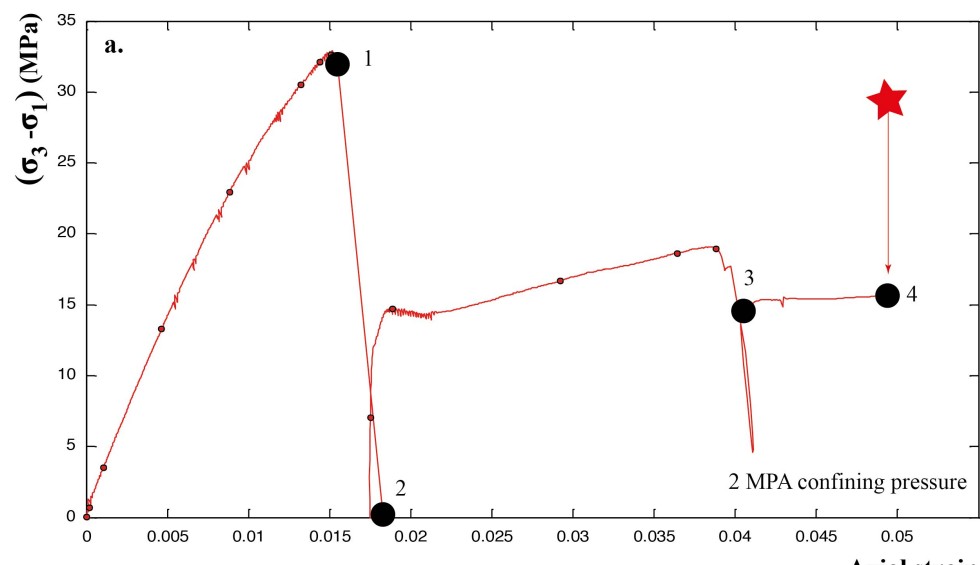

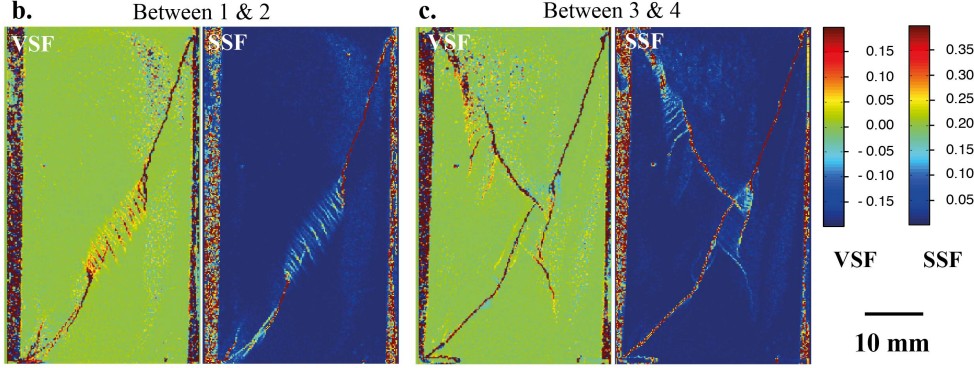

VSF: Volumetric strain field ; SSF: Maximum shear stress field

648                          **Figure 2**











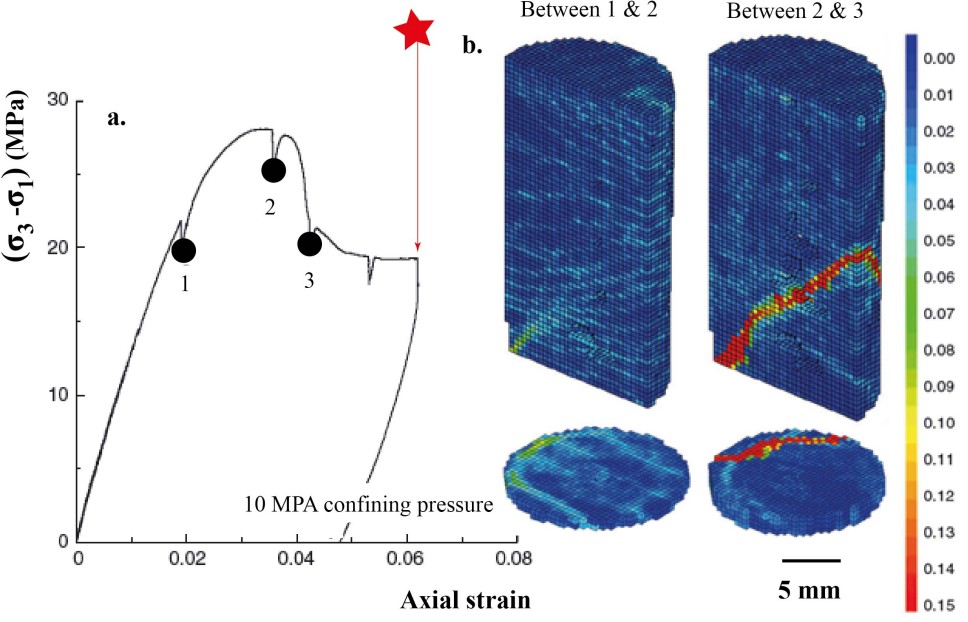


**Figure 3**










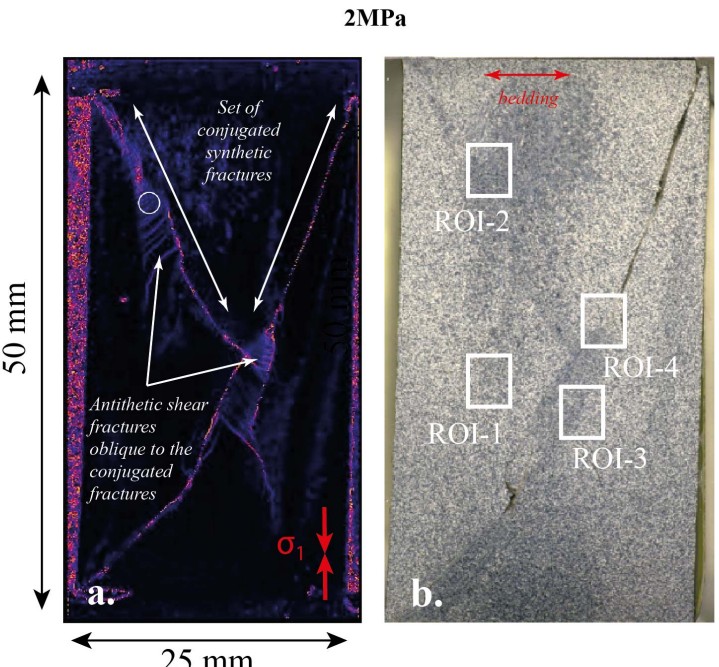

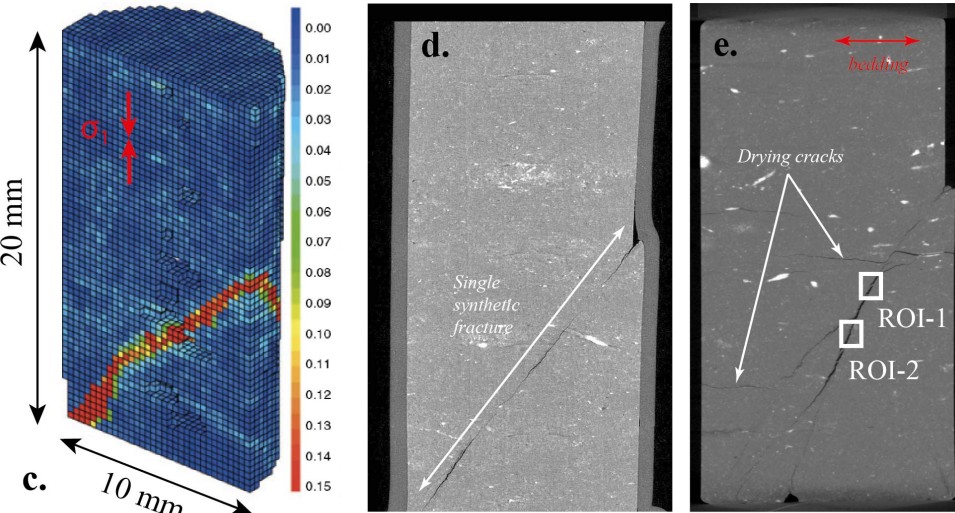

Figure 4






666                          **Figure 5**









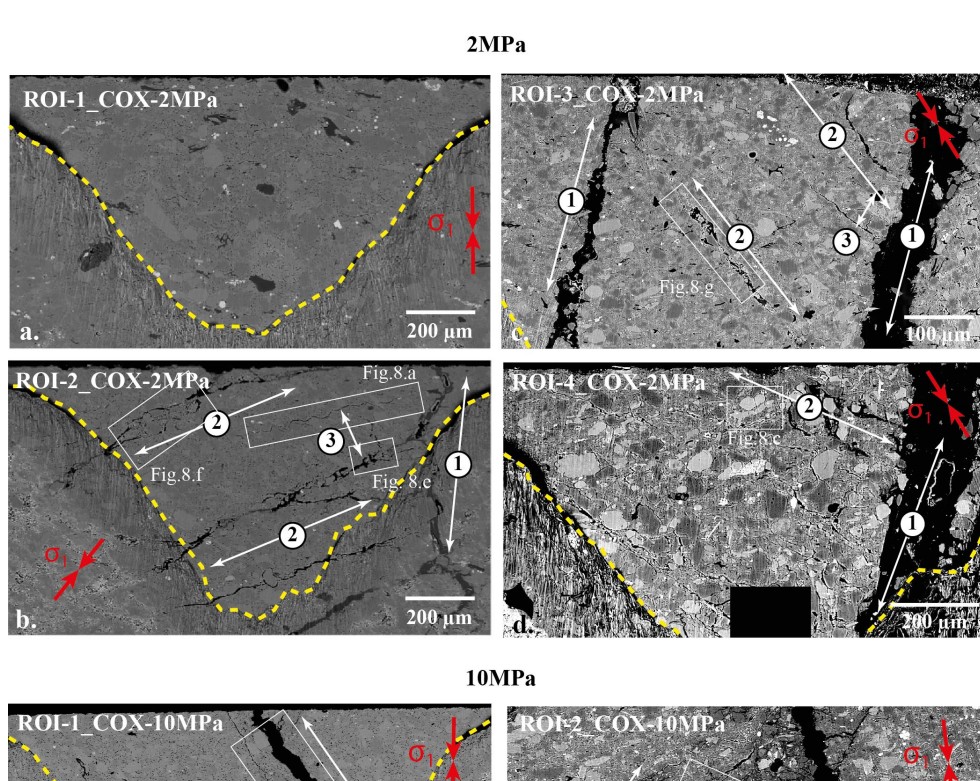

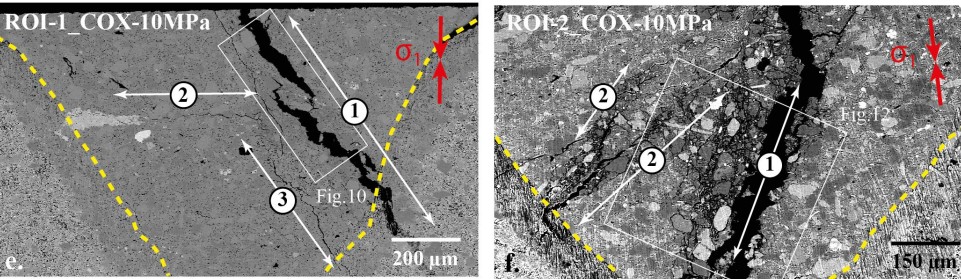


672                         **Figure 6**



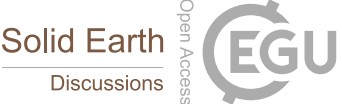







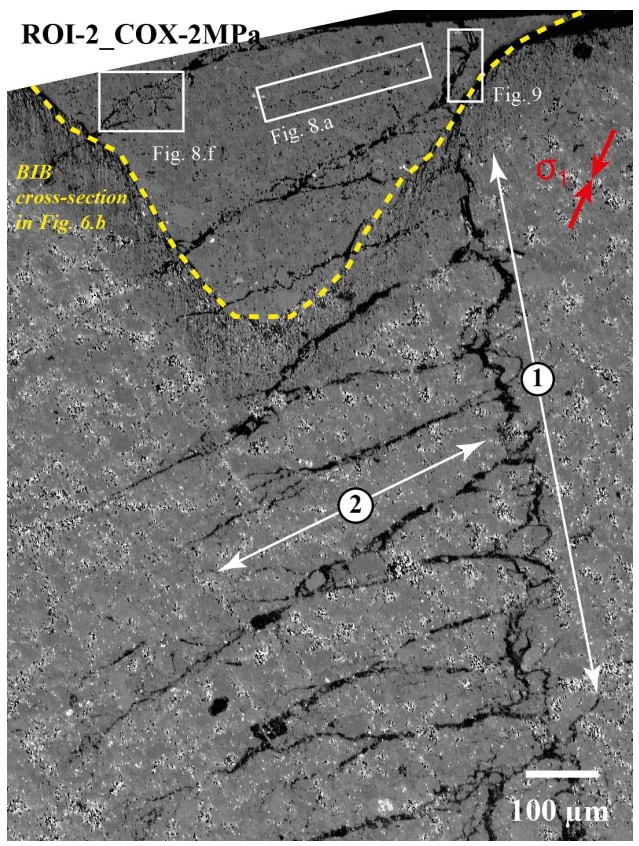



**Figure 7**





**Figure 8**







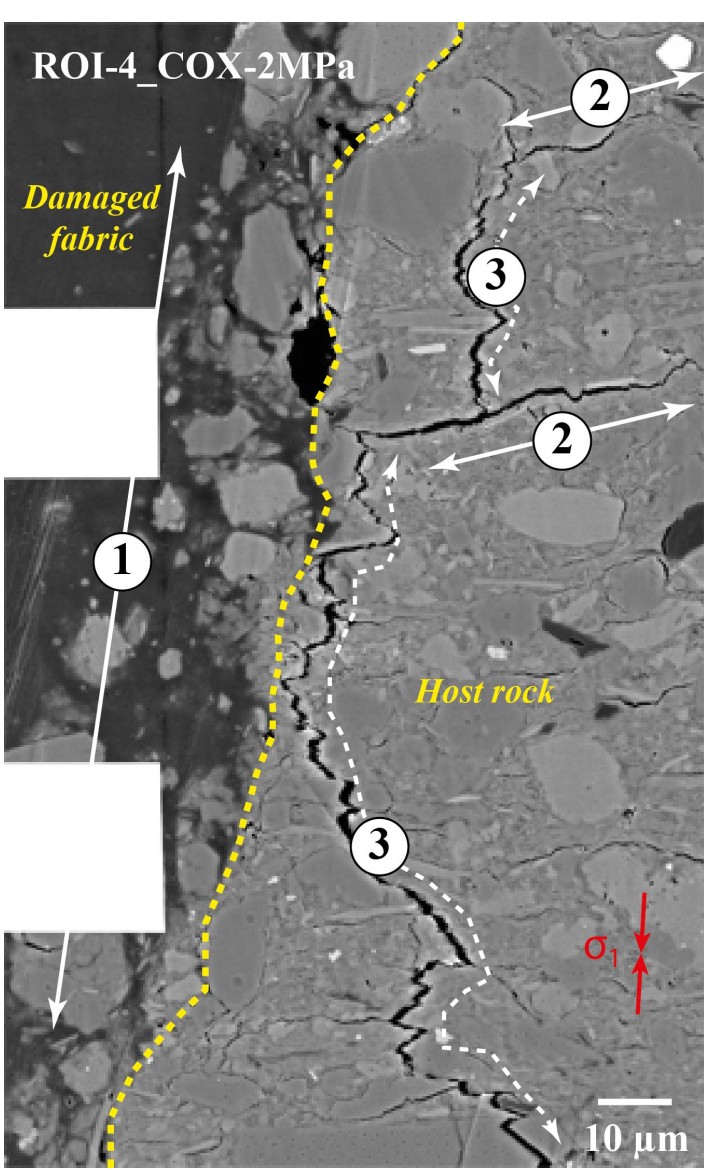


**Figure 9**







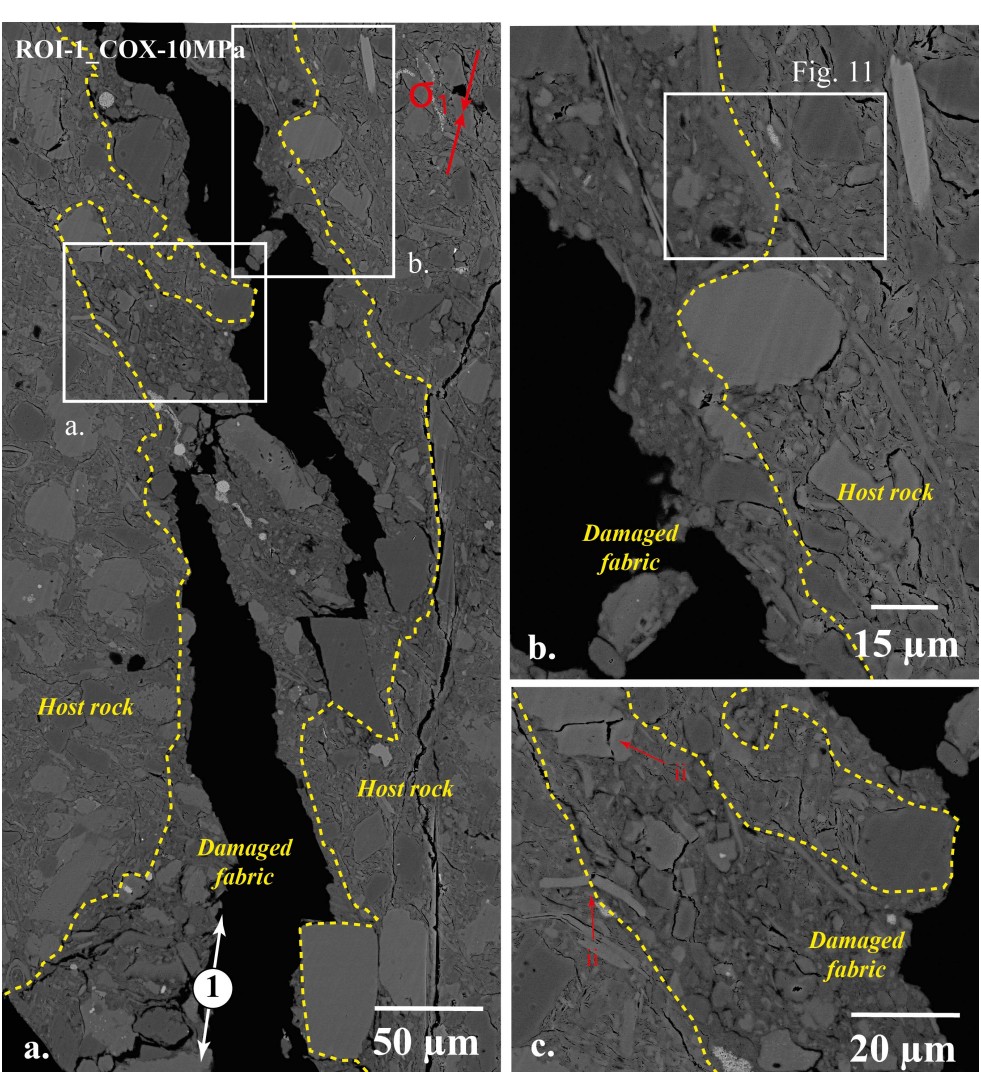


**Figure 10**








696                                    **Figure 11**






Figure 12


