# Peer review of "Cataclastic deformation of triaxially deformed, cemented mudrock"

_Solid Earth, 2016_

## Referee Comment (RC1) · A. Dimanov (Referee) · 7 Nov 2016

The paper "Cataclastic deformation of triaxially deformed, cemented mudrock (Cox Clay) : an experimental study at the micro/nano scale using BIB-SEM", by G. Desbois et al. is well written and well constructed, with a comprehensive rationale. It brings new insights on the deformation mechanisms active during experimental deformation of clay rich rocks from the deep underground (future) French nuclear waste repository. The work is carefully accomplished, thanks to edge cutting facilities (ion abrasion) allowing for preparation of high quality sample surfaces in order to access by SEM to the

fine scale microstructures and particularly down to the scale of the clay matrix. The work is based on techniques which are now well established (for instance, by the first author) and that have proven to be the most valuable for investigation of the microstructures of finely devised materials as clays. Similar approaches have been successfully applied by some of the authors to investigate porosity evolution and the mechanisms of damage in experimentally and naturally deformed clayey rocks. But, the deformed samples are always investigated at post mortem conditions. Therefore, the experimentalists do not always have access to the history of loading and only the final stage at failure provides guidance for the choice of the investigation areas. The improvement proposed in the present work is to select the investigated areas based on the in situ monitoring of deformation in samples deformed in previous studies, using digital image correlation, allowing for determining the full strain fields. The latter technique allows for instance to find out the localization of strain and damage in the samples during the loading process and therefore to seek the corresponding microstructures in the appropriate areas. I don't have any major problem with the philosophy of the experimental approach, nor with the organization of the paper and I recommend its publication. I have however few comments that follow: lines 69-75: Some rephrasing for clarity and paying attention to the tense may be needed. It should be clearly stated the different types of geomaterials (in addition to salt and clay-rocks, carbonates sould also be mentioned) which were tested and the type of in situ observation techniques (optical, SEM and X-ray tomography). Line 97: Yang et al (2012) used optical microscopy (not SEM). Line 113: I never heard about "crystal plasticity" of clays and think it is not appropriate to speak so. "Crystal plasticity" term may be misleading as it usually stands for crystal slip (dislocation glide) in massive crystalline materials, which is clearly not the structure of clay. Besides, it may suggest that something is already known about the "plasticity" mechanisms of clay particles, which is also not the case. Something is also mistaken in the phrasing : ". . . crystal plasticity of clay, a the poorly known plasticity of nano-clay aggregates. . .".* Line 164: check the figure, there is a mistake in the captions/ labelling of fig.2: it is written "maximum shear stress field", but DIC cannot measure stresses!

only strain! Also Fig. 3 repeats exactly a part of the synoptic figure 1, which small size makes it very difficult to read. It can probably be expanded and Fig.3 to be referred to this new Fig. 1, or something this way.

Similarly, Fig. 4 repeats the 3D strain field of the cylindrical sample already reported in Fig. 3. Some optimization in the presentation of these figures in order to avoid repeating several times the same elements would be appreciated. Line 247: Some precisions are needed. You state: "...fractures are not resolved by DIC". Yes, but this is only a question of 1) the resolution of the optical microscopy itself (camera, magnification, pixel size...), 2) the DIC local "strain gage length", or say the length scale of the marking contrasts and the specifically adopted procedure of calculation of strain from the displacement discrete field. Do not leave the reader with the impression that this is a general DIC limitation! Line 255: It must be clearly explained (probably well before this section) that the samples with 3D strain field measurements from Lenoir et al. were deformed in 2008! Since, we don't know how they were stored and preserved over nearly 10 years! This is what you probably call "slow drying", but state it more clearly and provide with more details about the way all the studied samples were stored; Line 290: The two previous DIC investigations can only indicate the local strain (compaction, shear, dilation...) at a given gage length, which is well above the inclusions size. Only your fine scale observations allow interpreting these strain fields in terms of mechanisms at the scale of the damaged inclusions. Anyway, you should recall the DIC "gage lengths". Line 313: "...3D and 2D digital image..." Finally, all my comments need only minor modifications and/or clarifications. All the best. A. Dimanov.

Please also note the supplement to this comment:
http://www.solid-earth-discuss.net/se-2016-131/se-2016-131-RC1-supplement.pdf

---

## Referee Comment (RC2) · G. Dresen (Referee) · 24 Nov 2016

Review of SE manuscript Cataclastic deformation of triaxially deformed, cemented mudrock (Callovo Oxfordian Clay): an experimental study at the micro/nano scale using BIB-SEM by Guillaume Desbois, Nadine Höhne, Janos L. Urai, Pierre Bésuelle, and Gioacchino Viggiani

The manuscript contains a detailed microstructural analysis of Bure clay samples that were previously subjected to different mechanical tests at confining pressures of 2 and 10 MPa. Sample deformation was recorded in situ using DIC and X-ray tomography,

respectively. Samples for the microstructure analysis presented here were carefully chosen with reference to the recorded deformation, and the analytical techniques used for this analysis are state of the art. The paper is generally well written and organized and could be published with minor revision. I have just a few comments listed below:

1. Section 3: Samples were extracted with a diamond saw and surfaces first polished using SiC paper and then BIB polished. Is this procedure sufficient to erase potential surface damage introduced during sawing and SiC polishing?

2. Section 4.1: It did not become clear to what extent and by which arguments the mode I fractures in either sample could be attributed to deformation or rather to unloading, drying etc. The authors refer to Figure 4d,e to illustrate the rather dramatic changes in the microstructure that occured over time. Should one not expect that most of the fractures that initially formed during deformation experienced some later overprint?

3. Section 4.2 and Discussion: Type II fractures show damage zones that are suggested to be wider in samples deformed at 10 MPa although porosity there is suppressed by shear-enhanced compaction. I would encourage the authors to elaborate on the micromechanisms forming the damage zones and involving cataclasis and pore collapse.

4. Section 4.2.2: Figure 8 is really busy and some arguments of the authors illustrated by this figure are hard to follow. For example, I find many chipped/angular non clay minerals also in the undeformed matrix (Figure 5, Figure 12a).

5. Figure 10: The epoxy impregnation indicating a damage zone is hard to see in this figure. Also, this is an image of a sample deformed at 10 MPa where porosity was suggested to be significantly reduced due to compaction. That would make it difficult for the epoxy to preferentially impregnate the damage zone, I would think.

6. Section 5.2: The authors consider the dominance of cataclastic deformation in these

samples surprising. Why? Differential stresses exceed the confining pressures by a factor of 3-15, which would suggest empirically that dilatatant fracturing prevails over other mechanisms (e.g. Kohlstedt et al., 1995).

I hope my comments are useful to the authors.

Sincerely

Georg Dresen

---

## Author Comment (AC1) · 1 Jan 2017

The paper "Cataclastic deformation of triaxially deformed, cemented mudrock (Cox Clay): an experimental study at the micro/nano scale using BIB-SEM", by G. Desbois et al. is well written and well constructed, with a comprehensive rationale. It brings new insights on the deformation mechanisms active during experimental deformation of clay rich rocks from the deep underground (future) French nuclear waste repository.

The work is carefully accomplished, thanks to edge cutting facilities (ion abrasion) allowing for preparation of high quality sample surfaces in order to access by SEM to the

fine scale microstructures and particularly down to the scale of the clay matrix. The work is based on techniques, which are now well established (for instance, by the first author) and that have proven to be the most valuable for investigation of the microstructures of finely devised materials as clays. Similar approaches have been successfully applied by some of the authors to investigate porosity evolution and the mechanisms of damage in experimentally and naturally deformed clayey rocks. But, the deformed samples are always investigated at post mortem conditions. Therefore, the experimentalists do not always have access to the history of loading and only the final stage at failure provides guidance for the choice of the investigation areas. The improvement proposed in the present work is to select the investigated areas based on the in situ monitoring of deformation in samples deformed in previous studies, using digital image correlation, allowing for determining the full strain fields. The latter technique allows for instance to find out the localization of strain and damage in the samples during the loading process and therefore to seek the corresponding microstructures in the appropriate areas.

I don't have any major problem with the philosophy of the experimental approach, nor with the organization of the paper and I recommend its publication.

I have however few comments that follow:

lines 69-75: Some rephrasing for clarity and paying attention to the tense may be needed. It should be clearly stated the different types of geomaterials (in addition to salt and clay-rocks, carbonates sould also be mentioned) which were tested and the type of in situ observation techniques (optical, SEM and X-ray tomography).

→ In the revised version of the manuscript, the method and the type of geomaterials are now indicated for each included reference. The text was also slightly reworded for better clarity.

Line 97: Yang et al (2012) used optical microscopy (not SEM). → Ok, we corrected this mistake in the revised version of the manuscript.

Line 113: I never heard about "crystal plasticity" of clays and think it is not appropriate to speak so. "Crystal plasticity" term may be misleading as it usually stands for crystal slip (dislocation glide) in massive crystalline materials, which is clearly not the structure of clay. Besides, it may suggest that something is already known about the "plasticity" mechanisms of clay particles, which is also not the case. Something is also mistaken in the phrasing : "...crystal plasticity of clay, a the poorly known plasticity of nano-clay aggregates...".

→ we agree only partly with the reviewer, "Crystal plasticity" is for sure not well known for phyllosilicates but "crystal plasticity of phyllosilicates" was already used in French et al. (2015), for example. By the way we reworded the related paragraph as following: "In summary, deformation mechanisms in mudrocks are poorly understood especially at low strain. Although as a first approximation the plasticity of cemented and uncemented mudrocks can be described by effective pressure- dependent constitutive models, the full description of their complex deformation and transport properties would be much improved by better understanding of the microscale deformation mechanisms. There is a wide range of possible mechanisms: intra- and intergranular fracturing, cataclasis, grain boundary sliding, grain rotation and granular flow, plasticity of phyllosilicates and the poorly known plasticity of nano-clay aggregates with the strong role of clay-bound water, cementation, fracture sealing and solution- precipitation."

French M.E., Chester F.M., Chester J.S. (2015). Micromechanisms of creep in clay-rich gouge from the Central Deforming Zone of the San Andreas Fault. Journal of geophysical research, 120, 827–849,

Line 164: check the figure, there is a mistake in the captions/ labelling of fig.2: it is written "maximum shear stress field", but DIC cannot measure stresses! only strain!

→ Ok, we corrected in the annotation of the figure 2

Also Fig. 3 repeats exactly a part of the synoptic figure 1, which small size makes it very difficult to read. It can probably be expanded and Fig.3 to be referred to this new Fig. 1, or something this way. Similarly, Fig. 4 repeats the 3D strain field of the cylindrical sample already reported in Fig. 3. Some optimization in the presentation of these figures in order to avoid repeating several times the same elements would be appreciated.

→ Ok, in the revised version of the manuscript, Figure 4 is deleted and Figure 2 and 3 updated including valuable information initially in Figure 4. Related figures captions are updated and figures references in the manuscript, also.

Line 247: Some precisions are needed. You state: "...fractures are not resolved by DIC". Yes, but this is only a question of 1) the resolution of the optical microscopy itself (camera, magnification, pixel size...), 2) the DIC local "strain gage length", or say the length scale of the marking contrasts and the specifically adopted procedure of calculation of strain from the displacement discrete field. Do not leave the reader with the impression that this is a general DIC limitation!

→ We included this remark in the revised version of the manuscript.

Line 255: It must be clearly explained (probably well before this section) that the samples with 3D strain field measurements from Lenoir et al. were deformed in 2008! Since, we don't know how they were stored and preserved over nearly 10 years! This is what you probably call "slow drying", but state it more clearly and provide with more details about the way all the studied samples were stored.

→ we give now this information much earlier (Section 3: Method) in the revised version of the manuscript. About the storage: samples were stored at low vacuum and room temperature in desiccator, where they dried slowly.

Line 290: The two previous DIC investigations can only indicate the local strain (compaction, shear, dilation...) at a given gage length, which is well above the inclusions size. Only your fine scale observations allow interpreting these strain fields in terms of mechanisms at the scale of the damaged inclusions. Anyway, you should recall the

DIC "gage lengths".

→ Ok, the DIC "gage length[s]" were recalled in the section 2 (material studied and DIC derived strain fields)

Line 313: "...3D and 2D digital image..."

→ Ok, we updated like this

Finally, all my comments need only minor modifications and/or clarifications.

All the best.

A. Dimanov

Please also note the supplement to this comment:
http://www.solid-earth-discuss.net/se-2016-131/se-2016-131-AC1-supplement.pdf

[Figure]

**Supplement:**

**Deformation in cemented mudrock (Callovo Oxfordian Clay) by micro-cracking, granular flow and phyllosilicate plasticity: insights from Triaxial Deformation, Broad ion Beam polishing and Scanning Electron Microscopy**

Guillaume Desbois[1], Nadine Höhne[1], Janos L. Urai[1], Pierre Bésuelle[2], Gioacchino Viggiani[3]

[1]*Structural Geology, Tectonics and Geomechanics, RWTH Aachen University, Lochnerstrasse 4-20, 52056 Aachen, Germany*

[2]*CNRS, 3SR, Grenoble, France*

[3]*Université Grenoble Alpes, 3SR, Grenoble, France*

**Abstract**

The macroscopic description of deformation and fluid flow in mudrocks can be improved by a better understanding of microphysical deformation mechanisms. Here we use a combination of Scanning Electron Microscopy (SEM) and Broad Ion Beam (BIB) polishing to study the evolution of microstructure in samples of triaxially deformed Callovo-Oxfordian Clay. Digital Image Correlation (DIC) was used to measure strain field in the samples, and as a guide to select regions of interest in the sample for BIB-SEM analysis. Microstructures show evidence for dominantly cataclastic and minor crystal plastic mechanisms (intergranular, transgranular, intragranular cracking, grain rotation, clay particle bending) down to nm- scale. At low strain, the dilatant fabric contains individually recognizable open fractures, while at high strain the reworked clay gouge also contains broken non-clay grains and smaller pores than the undeformed material, resealing the initial fracture porosity.
**Keywords:** cemented mudrock, Callovo Oxfordian Clay, triaxial deformation, clay microstructure, deformation mechanisms, BIB-SEM, DIC, cataclastic deformation

**1   Introduction**

Mudrocks constitute up to 80% of the Earth's sedimentary rocks (Stow, 1981). Due to their low permeability and self-sealing properties (Boisson, 2005, Bernier et al., 2007), claystones are considered for nuclear waste disposal and seals for storage in deep geological formations (Salters & Verhoef, 1980; Shapira 1989; Neerdael & Booyazis, 1997; Bonin, 1998; Ingram & Urai, 1999; ONDRAF/NIRAS, 2001; NAGRA, 2002; NEA, 2004; ANDRA 2005; IAEA, 2008). Predictions of mechanical and transport properties over long time scales are essential for the evaluation of subsurface integrity. For this, it is generally agreed that a multiscale experimental approach that combines measurement of bulk mechanical and transport properties with microstructural study to identify deformation mechanisms is required to develop microphysics-based constitutive equations, which can be extrapolated to time scales not available in the laboratory, after comparison with naturally deformed specimens (Morgenstern & Tchalenko 1967; Tchalenko, 1968; Lupini et al., 1981; Rutter et al., 1986; Logan et al., 1979, 1987, 1992; Marone & Scholz, 1989; Evans & Wong, 1992; ; Katz &

Reches, 2004; Niemeijer & Spiers, 2006. Colletini et al., 2009; Haines et al., 2009, 2013; French et al.,

2015; Crider, 2015; Ishi, 2016).

In the field of rock mechanics and rock engineering, experiments are performed to low strain and over relatively short time in order to predict damage and deformation in tunnelling and mining, for example.

Here, a macroscopic and phenomenological approach is common to characterize mechanical and transport properties and to establish the constitutive laws. Microstructures are rarely studied because the strained regions are difficult to find (except macroscopic fractures), and because microstructures below micrometre scales are elusive. However, it is well established that for long-term predictions a microphysics-based understanding of mechanical and fluid flow properties in mudrocks provides a better basis for extrapolating constitutive equations beyond the time scales accessible in the laboratory.

This requires integration of measurement of the mechanical and transport properties with microstructures, towards multi-scale description of deformation in mudrocks at low strain.

The microstructural geology community studied microstructures in deformed mudrocks to infer deformation mechanisms (Dehandschutter et al., 2004; Gratier et al., 2004; Klinkenberg et al; 2009;

Renard, 2012; Robinet et al., 2012; Richard et al., 2015; Kaufhold et al., 2016), but this was limited by problems with sample preparation for high resolution electron microscopy. On the other hand, the mechanical properties and related microstructures of natural and experimental high strain fault rocks have been studied extensively (Bos & Spiers, 2001; Faulkner et al., 2003, Marone & Scholz, 1989).

For Opalinus Clay (OPA) deformed in laboratory, Nüesch (1991) and Jordan and Nüesch (1989)

concluded that cataclastic flow was the main deformation mechanism, with kinking and shearing on

R- and P-surfaces at the micro scale, however this was only based on observations with optical microscopy, so that grain scale processes were not resolved. Klinkerberg et al. (2009) demonstrated a correlation between compressive strength and carbonate content of two claystones; this correlation is positive for OPA but negative for Callovo Oxfordian Clays (COX). This was explained by the differences in grain size, shape, and spatial distribution of the carbonate (Klinkerberg et al. 2009), cf.

Bauer-Plaindoux et al. (1998).  Microstructural investigations using BIB-SEM and FIB-TEM in OPA

from the main fault in the Mt-Terri Underground Research Laboratory (Laurich et al., 2014, 2016)

showed that inter- and transgranular microcracking, pressure solution, clay neoformation, phyllosilicate crystal plasticity and grain boundary sliding all play an important role during the early
stages of faulting in OPA. However, simple cataclastic microstructures are rare due to the high shear
strain and there was an almost complete loss of porosity in micro- shear zones.

Digital Image Correlation (DIC) applied on images acquired during experimental deformation
provides a method to measure directly the local displacement fields (in 2D or 3D depending on the
imaging method) and quantifies locally strain over time (Lenoir et al., 2007 [Claystone, 3D, X-ray
tomography]; Bornert et al., 2010 [Claystone, 2D, optical microscopy]; Bésuelle & Hall, 2011
[Claystone, 2D, Optical microscopy]; Dautriat et al., 2011 [Carbonates, 2D, optical microscopy and
SEM]; Wang et al., 2013, 2015 [Claystone, 2D, environmental SEM]; Fauchille et al., 2015
[Claystone, 2D, Optical microscopy]; Sone et al., 2015 [Shale, 2D, SEM]). For samples with grain
sizes above micrometres, this approach allows studying processes occurring at grain scale with high
resolution (Hall et al., 2010 [Sand, 3D, X-ray tomography]; Andò et al., 2012 [Sand, 3D, X-ray
tomography]; Bourcier et al., 2012, 2013 [rock salt, 2D, optical microscopy and environmental SEM];
Wang et al., 2015 [Claystone, 2D, environmental SEM]). On claystones, DIC was used to study
swelling in environmental SEM (Wang et al., 2013, 2015) to measure strain between the clay matrix
and non-clay minerals.

Microstructural studies in naturally compacted mudrocks are currently in rapid development, enabled
by the development of ion beam milling tools (e.g. Focussed Ion Beam [FIB] and Broad Argon Ion
Beam [BIB]) which allow imaging of mineral fabrics and porosity down to nm- scale in very high
quality cross sections with SEM and TEM (Lee et al., 2003; Desbois et al., 2009, 2011, 2013, 2016;
Loucks et al., 2009; Curtis et al., 2010; Heath et al., 2011; Klaver et al., 2012; Keller et al., 2011, 2013;
Houben et al., 2013, 2014; Hemes et al., 2013, 2015; Laurich et al., 2014; Warr et al., 2014; Song et
al., 2016). Serial sectioning allows reconstruction of microstructure in 3D (Keller et al., 2011, 2013;
Milliken et al., 2013; Hemes et al., 2015), and cryogenic techniques can image the pore fluid in the
samples and avoid artefacts produced by drying (Desbois et al., 2013, 2014; Schmatz et al., 2015).

Previous work has shown that the mechanical properties of Callovo Oxfordian Clays (COX) do not
only depend on the fraction and mineralogy of the clay but also on water content and texture (Bauer-
Plaindoux et al., 1998). Chiarelli et al. (2000) showed that COX is more brittle with increasing calcite
content and more ductile with increasing clay content and proposed two deformation mechanisms:
plasticity induced by slip of clay sheets and induced anisotropic damage as indicated by microcracks
at the interface between grains and matrix, however they provided little microstructural evidence to
support this. Gasc-Barbier et al., (2004), Fabre et al., (2006), Chiarelli et al., (2003), Fouché et al.,
(2004) report that the COX has an unconfined compressive strength of 20 to 30 MPa and a Young's
modulus of 2 to 5 GPa. In the context of underground storage of radioactive wastes, these papers try to
predict the mechanical evolution of COX over the period of thousands of years. The effects studied include creep, pore-pressure dissipation, swelling, contraction, chemical effects, pressure solution and
force of crystallization. Although these papers develop elaborate constitutive laws, they provide very
limited microstructural observations. The need for micromechanical observations was already
recognized by Yang et al. (2012) and Wang et al., (2013, 2015). From Digital Image Correlation
(DIC) applied on optical and ESEM images, these authors have shown how heterogeneous strain fields
correlate with microstructure and recognized shear bands and tensile microcracks.

For highly overconsolidated claystones from the Variscan foreland thrust belt in the Ardennes and
Eifel, Holland et al. (2006) proposed an evolutionary model starting with mechanical fragmentation of
the original fabric. In this model, the initial loss of cohesion is driven by kinking, folding and micro-
fracturing processes with an increasing porosity and permeability. Abrasion during progressive
deformation increases the amount of clay gouge, and re-sealing occurs by decrease in pore size of the
clay gouge.

In summary, deformation mechanisms in mudrocks are poorly understood especially at low strain.
Although as a first approximation the plasticity of cemented and uncemented mudrocks can be
described by effective pressure- dependent constitutive models, the full description of their complex
deformation and transport properties would be much improved by better understanding of the
microscale deformation mechanisms. There is a wide range of possible mechanisms: intra- and
intergranular fracturing, cataclasis, grain boundary sliding, grain rotation and granular flow, plasticity
of phyllosilicates and the poorly known plasticity of nano-clay aggregates with the strong role of clay-
bound water, cementation, fracture sealing and solution- precipitation.

[revised manuscript text omitted]

**181   3    Methods: BIB-SEM imaging of deformed microstructures**

After the experiments of Lenoir et al. (2007) and Bésuelle et Hall (2011), deformed samples were stored at low vacuum and room temperature in a desiccator, where they dried slowly. From these deformed samples, sub-samples were selected to represent areas with different strain history based on the DIC analysis. For COX-2MPa, three BIB cross sections were prepared around the conjugate fractures in areas with different amount of diffuse strain (at the resolution of DIC), antithetic fractures (ROI-2, ROI-3 and ROI-4; Figures 2.d, 5.b, c, d and 6) and a fourth one in a region without measurable strain (ROI-1; Figures 2.d and 5.a). For COX-10MPa, two BIB-SEM analyses were done around the single shear fracture (Figures 3.d and 5.e, f).

Sub-samples were first embedded in epoxy, extracted with a low speed diamond saw in dry conditions, pre-polished dry using SiC papers (down to P4000 grade) and BIB polished in a JEOL

SM-09010 cross-section polisher (for 8 h, $1.10^{-3} – 1.10^{-4}$ Pa, 6 kV, 150 μA) to remove a 100 μm thick layer of material interpreted to be the layer of damage after polishing with SiC papers. BIB cross- sections are all prepared parallel to the $\sigma_1$ and direction and perpendicular to the shear fracture. The

[revised manuscript text omitted]

Cosenza et al., 2007; Pineda et al., 2010; Hedan et al., 2012; Renard, 2012; Wang et al., 2013, 2015;
Desbois et al., 2014).

The DIC analysis is not affected by this because the images were acquired during deformation of
preserved (wet) samples. SEM analysis is done on samples which have been deformed and unloaded,
followed by slow drying in low vacuum and further dehydration in the high vacuum of the BIB and
SEM. In COX-10MPa, this is illustrated by Figures 3.c and 3.d. Figure 3.c shows the sample at the
end of the deformation experiment, whereas Figure 3.d shows the same sample but about 10 years
later, both imaged with X-ray. The comparison of Figures 3.c and 3.d shows that cracks developed
parallel to the bedding and that the apertures of fractures developed during the deformation became
larger. These are interpreted to result from unloading and shrinkage during drying of specimens.
Though the second sample was not scanned with X-ray in dry condition, we infer that similar changes
occurred also in COX-2MPa: by analogy, there is no reason that the clay matrix in COX-2MPa
behaves differently that in COX-10MPa.

The considerations above indicate that some fractures developed during deformation but drying
damage overprinted them. Unfortunately, BIB-SEM images (performed on dried samples) do not
provide direct information to distinguish if the visible fractures and cracks developed during
deformation (and subsequently overprinted by drying) or only by drying. However, as presented in the
following paragraphs, indirect evidence suggests that the fractures in the fragments between the arrays
of antithetic fractures and the antithetic fractures of *Type I* and of *Type II* developed during
deformation.

[revised manuscript text omitted]

**5.3 Conceptual model of microstructure development in triaxially deformed COX.**

Based on BIB-SEM microstructural observations, we propose the following sequence of micro-
mechanisms Callovo-Oxfordian clay (Figure 12):

(1) & (2) Micro-fracturing

Incipient deformation occurs by intergranular microfractures propagating in the clay matrix and,
transgranular and intragranular micro fractures in non-clay minerals, both resulting in the
fragmentation of the original fabric and in agreement with the high compressive strength of this
cemented mudstone. Intergranular micro fractures are interpreted to be initiated from pores,
propagating along weak contacts at non-clay mineral / clay matrix interfaces or along (001) cleavage
planes of phyllosilicates (Chiarelli et al., 2000; Klinkenberg et al., 2009; Den Hartog & Spiers, 2014,
Jessel et al., 2009). Here note that probably the biggest unknown at present in the micro-mechanisms
of deformation in claystones is the nature of cement bonds between grains; further work in this project
is aimed at understanding this better.

(3 & 4) Cataclastic shearing with plasticity of phyllosilicates, macroscopic failure

Further deformation occurs by frictional sliding affecting the process zone at microfracture boundaries,
and in relays between fractures. Mechanisms are abrasion and bending of phyllosilicates by cataclastic
and crystal plastic mechanisms. This is accompanied by rotation of fragments and cataclastic flow.
This stage is interpreted to start at the peak stress in the stress-strain curve, accompanied by local
dilatancy. At the specimen scale, fractures link up resulting in loss of cohesion. In restraining sections
along the fractures, reworking of the clay matrix reduces porosity and eliminates large pores, changing
the pore size distributions. The specimen suffers from a major loss of cohesion accompanied by
dilatancy and stress drop after peak stress.

(5) Resealing of the damage zone by shear and pore collapse, evolution of clay gouge

Ongoing abrasion of the fragments and comminution develop a cataclastic fabric. A full understanding
of the deformation mechanisms in cataclastic clay aggregates requires more work, but the grain sliding
(Chiarelli et al., 2000) and grain rotation between low-friction clay particles together with collapsing
of porosity is inferred because: (i) slip on the (001) basal planes of clay particles is much easier than shearing related to grain breakage (cf. Haines et al., 2013 and Crider, 2015) and (ii) residual strength observed after specimen's failure argues for sliding between low frictional clay particles (Lupini et al., 1981). At sufficently high strain this stage would correspond to the residual strength result in the resealing of initial fracture porosity by filling the fractures with clay gouge. In this stage, cataclasis of non-clay particles is expected to become less important because they are embedded in reworked clay.

The conceptual model above for microstructure evolution in triaxially deformed COX is a first look based on direct grain-scale observation of microstructures. Our ongoing studies focus on the nature of the cement and at microstructures of the damage zone at fracture tips to better understand the localization mechanisms.

**6   Conclusions**

[revised manuscript text omitted]

Ishii, E., H. Sanada, H. Funaki, Y. Sugita, and H. Kurikami (2011), The relationships among brittleness, deformation behavior, and transport properties in mudstones: An example from the Horonobe Underground Research Laboratory, Japan, J. Geophys. Res., 116, B09206, doi:10.1029/2011JB008279.

Jessell, M.W., Bons, P.D., Griera, A., Evans, L. & Wilson, C.J.L. 2009. A tale of two viscosities. Journal of Structural Geology, 31: 719-736.s

Jordan P. and Nüesch R. (1989) Deformation behavior of shale interlayers in evaporite detachment horizons, Jura overthrust, Switzerland. Journal of Structural Geology, 11(7): 859-871.

Kang M-S., Watabe Y. and Tsuchida T. (2003). Effect of Drying Process on the Evaluation of Microstructure of Clays using Scanning Electron Microscope (SEM) and Mercury Intrusion Porosimetry (MIP). Proceedings of The Thirteenth (2003) International Offshore and Polar Engineering Conference Honolulu, Hawaii, USA, May 25–30, 2003

Katz O. and Reches Z. (2004). Microfracturing, damage, and failure of brittle granites. Journal of Geophysical Research, 109, B01206.

Kaufhold A., Halisch M., Zacher G., Kaufhold S. (2016). X-ray computed tomography investigation of structures in Opalinus Clay from large-scale to small-scale after mechanical testing. Solid Earth, 7: 1171-1183.

Keller L., Schuetz P., Erni R., Rossell, M.D., Lucas, F., Gasser, Ph., Holzer L. (2013). Characterization of multi-scale microstructural features in Opalinus Clay. Microporous and Mesoporous Materials, 170 : 83-94.

Keller L.M., Holzer L., Wepf R., Gasser P. (2011). 3D geometry and topology of pore pathways in Opalinus clay: Implications for mass transport. Applied Clay Science 52: 85-95.

Kim Y-S, Peacock D.C.P, Sanderson D.J. (2004). Fault damage zones. Journal of Structural Geology, 26: 503-517.

Klaver J., Desbois G., Littke R., Urai J.L. (2015). BIB-SEM characterization of pore space morphology and distribution in postmature to overmature samples from the Haynesville and Bossier Shales, Marine and Petroleum Geology, 59: 451-466.

Klaver J., Desbois G., Urai J.L. and Littke R. (2012). BIB-SEM study of porosity of immature Posidonia shale from the Hils area, Germany. International Journal of Coal Geology, 103: 12-25.

Klinkenberg M., Kaufhold S., Dohrmann R., Siegesmund S. (2009). Influence of carbonate microfabrics on the failure strength of claystones. Engineering Geology 107: 42-54.

Kohlstedt D.L., Evans B., Mackwell S.J. (1995). Strength of the lithosphere: constraints imposed by laboratory experiments. Journal of geophysical Research, 100(B9): 17587-17602.

Laurich B., Urai J.L., Desbois G., Vollmer C., Nussbaum C. (2014). Microstructural evolution of an incipient fault zone in Opalinus Clay: Insights from an optical and electron microscopic study of ion-beam polished samples from the Main Fault in the Mont Terri underground research laboratory. Journal of Structural Geology,
67: 107–128.

Laurich B., Urai J.L., Nussbaum C. (2016). Microstructures and deformation mechanisms in Opalinus Clay:
insights from scaly clay from the Main Fault in the Mont Terri Rock Laboratory (CH).Solid Earth,
doi:10.5194/se-2016-94

[revised manuscript text omitted]

Lenoir et al., 2007;).

**Figure 2:** Results of deformation test done on sample COX-2MPa. (a): deviator stress vs. axial strain response. The red star indicates the state of sample when BIB-SEM microstructural analyses are done.

(b) and (c): incremental volumetric strain fields (VSF) and maximum shear strain fields (SSF) fields for deformation increment 1-2 and 3-4 indicated in (a) after DIC. Arrows with solid lines indicate the set of two conjugated synthetic fractures whereas the arrows with dashed lines show antithetic fractures oblique to the conjugated fractures. (d): Selection of differently strained areas (ROI)

highlighted from DIC analysis for BIB-SEM microstructural analyses. Four ROI were analysed: three at conjugate synthetic fractures in areas with different amount of diffuse strain and antithetic fractures (ROI-2, ROI-3 and ROI-4) and one in a region without measurable strain (ROI-1). After Bésuelle et al.

(2011).

**Figure 3:** Results of deformation test done on sample COX-10MPa. (a): deviator stress vs. axial strain response. The red star indicates the state of sample when BIB-SEM microstructural analyses are done.

(b): incremental maximum shear strain fields for deformation increment 1-2 and 2-3 indicated in (a)

interpreted after DIC.  (c) shows the X-ray radiography of the sample taken directly at the end of the deformation test, whereas (d) shows the X-ray radiography of the same sample but taken about 10

years after the end of the deformation: drying cracks developed following the bedding and the aperture of the single shear fracture became larger. (d) indicates also two ROI were analysed both around the single synthetic shear fracture. In (c) and (d): orientation of $s_1$ and of the bedding are indicated in red.

After Lenoir et al. (2007).

[revised manuscript text omitted]

.

**Figures**

[Figure]

**Figure 1**

[Figure]

**Figure 2**

[Figure]

**Figure 3**

[Figure]

[Figure]

**Figure 4**

[Figure]

Figure 5

**Figure 5**

[Figure]

**Figure 6**

[Figure]

**Figure 7**

[Figure]

**Figure 8**

[Figure]

a. ROI-1 COX-10MPa
σ₁
b.
Host rock
Host rock
Damaged fabric
µm
a.

b. Fig. 11
Host rock
Damaged fabric
µm c. ii
ii
Damaged fabric
µm

**Figure 9**

[Figure]

**Figure 10**

[Figure]

**Figure 11**

[Figure]

**Figure 12**

---

## Author Comment (AC2) · 1 Jan 2017

Review of SE manuscript Cataclastic deformation of triaxially deformed, cemented mudrock (Callovo Oxfordian Clay): an experimental study at the micro/nano scale using BIB-SEM by Guillaume Desbois, Nadine Höhne, Janos L. Urai, Pierre Bésuelle, and Gioacchino Viggiani.

The manuscript contains a detailed microstructural analysis of Bure clay samples that were previously subjected to different mechanical tests at confining pressures of 2 and 10 MPa. Sample deformation was recorded in situ using DIC and X-ray tomography, respectively. Samples for the microstructure analysis presented here were carefully chosen with reference to the recorded deformation, and the analytical techniques used for this analysis are state of the art. The paper is generally well written and organized and could be published with minor revision. I have just a few comments listed below:

1. Section 3: Samples were extracted with a diamond saw and surfaces first polished using SiC paper and then BIB polished. Is this procedure sufficient to erase potential surface damage introduced during sawing and SiC polishing?

$\rightarrow$ After extraction of sub-samples by sawing, we pre-polished manually sub-samples with SiC grinding paper down to a grade of P4000 (i.e. a median grain size of 2.5 $\mu$m), removing damage from sawing. Subsequently, we used BIB milling to prepare high quality polished cross-section. The BIB cross-sectioner removes about 100 $\mu$m of the SiC milled surface. Therefore, BIB cross sectioning erases all potential surface damage introduced during SiC polishing. This information is now specified in the reviewed version of the manuscript (Lines 287 to 288 in the revised version).

2. Section 4.1: It did not become clear to what extent and by which arguments the mode I fractures in either sample could be attributed to deformation or rather to unloading, drying etc. The authors refer to Figure 4d,e to illustrate the rather dramatic changes in the microstructure that occured over time. Should one not expect that most of the fractures that initially formed during deformation experienced some later overprint?

$\rightarrow$ We agree with the reviewer: some fractures were initially formed during deformation (Fig. 4.d) and later overprinted by drying (Fig. 4.e). Here, drying tends to make the aperture of syn-deformation fractures larger. Unfortunately, BIB-SEM images (performed on dried samples) do not provide direct information to know if the visible fractures and cracks developed during deformation (and subsequently overprinted by drying) or only by drying. However, as presented in the second part of the Section 5.1, indirect evidences argue that the fractures in the fragments between the arrays of an- tithetic fractures and the antithetic fractures of Type I and of Type II developed during deformation. In the reviewed version of the manuscript, we updated the Section 5.1 (in discussion) to make it clearer.

3. Section 4.2 and Discussion: Type II fractures show damage zones that are suggested to be wider in samples deformed at 10 MPa although porosity there is suppressed by shear-enhanced compaction. I would encourage the authors to elaborate on the micromechanisms forming the damage zones and involving cataclasis and pore collapse.

→ In the revised version of the manuscript we added a new section (Section 5.3, in Discussion) titled " Conceptual model of microstructure development in triaxially deformed COX".

4. Section 4.2.2: Figure 8 is really busy and some arguments of the authors illustrated by this figure are hard to follow. For example, I find many chipped/angular non clay minerals also in the undeformed matrix (Figure 5, Figure 12a).

→ We find that Figure 8 is not too busy but we can split it into 2 figures if the editor thinks it is necessary. There are chipped/angular non-clay minerals also in the undeformed matrix. This is true but not "many" as suggested by the reviewer. In comparison, the damage zone is built with "many" of broken grains/fragments (e.g. Figure 12). In the revised version of the manuscript, we reworded slightly the section 4.2.2 to clarify this above.

5. Figure 10: The epoxy impregnation indicating a damage zone is hard to see in this figure. Also, this is an image of a sample deformed at 10 MPa where porosity was suggested to be significantly reduced due to compaction. That would make it difficult for the epoxy to preferentially impregnate the damage zone, I would think.

→ We think that the reviewer refers to the lines 233-234 in the original version of the manuscript. In this case, we fully agree about the comments above. The reference to the Figure 10 in lines 233-234 is a mistake. We deleted the reference to the figure 10 in lines 233-234 in the revised version of the manuscript.

6. Section 5.2: The authors consider the dominance of cataclastic deformation in these samples surprising. Why? Differential stresses exceed the confining pressures by a factor of 3-15, which would suggest empirically that dilatatant fracturing prevails over other mechanisms (e.g. Kohlstedt et al., 1995).

→ Reviewer has right, cataclastic deformation is not "surprising". Therefore we updated the first paragraph of the section 5.2 as following:

"In our experiments, differential stresses exceed the confining pressure by a factor of 3-15, which would suggest that dilatant fracturing prevails over other mechanisms (e.g. Kohlstedt et al., 1995). This is partly corroborated from the stress-strain measurements that show major stress drops after peaks of stress (Figures 2 and 3). In agreement with this, at micro-scale the first conclusion based on the microstructural observations above is the dominantly cataclastic deformation in Callovo-Oxfordian Clay at confining pressures up to 10 MPa. Microfracturing, producing fragments at a range of scales and reworking into a phyllosilicate-rich cataclastic gouge during frictional flow are the main processes in both samples. This is accompanied by dilatancy and by microfracturing of the original fabric, but also by progressive decrease of porosity and pore size in the gouge with the non-clay particles embedded in reworked clay. The structure of macro-scale fracture in the samples compares well with Ishii et al., (2011, 2016). "

I hope my comments are useful to the authors. Sincerely

Georg Dresen

Please also note the supplement to this comment:
http://www.solid-earth-discuss.net/se-2016-131/se-2016-131-AC2-supplement.pdf

[Figure]

**Supplement:**

**Deformation in cemented mudrock (Callovo Oxfordian Clay) by micro-cracking, granular flow and phyllosilicate plasticity: insights from Triaxial Deformation, Broad ion Beam polishing and Scanning Electron Microscopy**

Guillaume Desbois[1], Nadine Höhne[1], Janos L. Urai[1], Pierre Bésuelle[2], Gioacchino Viggiani[3]

[1]*Structural Geology, Tectonics and Geomechanics, RWTH Aachen University, Lochnerstrasse 4-20, 52056 Aachen, Germany*

[2]*CNRS, 3SR, Grenoble, France*

[3]*Université Grenoble Alpes, 3SR, Grenoble, France*

**Abstract**

The macroscopic description of deformation and fluid flow in mudrocks can be improved by a better understanding of microphysical deformation mechanisms. Here we use a combination of Scanning Electron Microscopy (SEM) and Broad Ion Beam (BIB) polishing to study the evolution of microstructure in samples of triaxially deformed Callovo-Oxfordian Clay. Digital Image Correlation (DIC) was used to measure strain field in the samples, and as a guide to select regions of interest in the sample for BIB-SEM analysis. Microstructures show evidence for dominantly cataclastic and minor crystal plastic mechanisms (intergranular, transgranular, intragranular cracking, grain rotation, clay particle bending) down to nm- scale. At low strain, the dilatant fabric contains individually recognizable open fractures, while at high strain the reworked clay gouge also contains broken non-clay grains and smaller pores than the undeformed material, resealing the initial fracture porosity.
**Keywords:** cemented mudrock, Callovo Oxfordian Clay, triaxial deformation, clay microstructure, deformation mechanisms, BIB-SEM, DIC, cataclastic deformation

**1   Introduction**

Mudrocks constitute up to 80% of the Earth's sedimentary rocks (Stow, 1981). Due to their low permeability and self-sealing properties (Boisson, 2005, Bernier et al., 2007), claystones are considered for nuclear waste disposal and seals for storage in deep geological formations (Salters & Verhoef, 1980; Shapira 1989; Neerdael & Booyazis, 1997; Bonin, 1998; Ingram & Urai, 1999; ONDRAF/NIRAS, 2001; NAGRA, 2002; NEA, 2004; ANDRA 2005; IAEA, 2008). Predictions of mechanical and transport properties over long time scales are essential for the evaluation of subsurface integrity. For this, it is generally agreed that a multiscale experimental approach that combines measurement of bulk mechanical and transport properties with microstructural study to identify deformation mechanisms is required to develop microphysics-based constitutive equations, which can be extrapolated to time scales not available in the laboratory, after comparison with naturally deformed specimens (Morgenstern & Tchalenko 1967; Tchalenko, 1968; Lupini et al., 1981; Rutter et al., 1986; Logan et al., 1979, 1987, 1992; Marone & Scholz, 1989; Evans & Wong, 1992; ; Katz &

Reches, 2004; Niemeijer & Spiers, 2006. Colletini et al., 2009; Haines et al., 2009, 2013; French et al.,

2015; Crider, 2015; Ishi, 2016).

In the field of rock mechanics and rock engineering, experiments are performed to low strain and over relatively short time in order to predict damage and deformation in tunnelling and mining, for example.

Here, a macroscopic and phenomenological approach is common to characterize mechanical and transport properties and to establish the constitutive laws. Microstructures are rarely studied because the strained regions are difficult to find (except macroscopic fractures), and because microstructures below micrometre scales are elusive. However, it is well established that for long-term predictions a microphysics-based understanding of mechanical and fluid flow properties in mudrocks provides a better basis for extrapolating constitutive equations beyond the time scales accessible in the laboratory.

This requires integration of measurement of the mechanical and transport properties with microstructures, towards multi-scale description of deformation in mudrocks at low strain.

The microstructural geology community studied microstructures in deformed mudrocks to infer deformation mechanisms (Dehandschutter et al., 2004; Gratier et al., 2004; Klinkenberg et al; 2009;

Renard, 2012; Robinet et al., 2012; Richard et al., 2015; Kaufhold et al., 2016), but this was limited by problems with sample preparation for high resolution electron microscopy. On the other hand, the mechanical properties and related microstructures of natural and experimental high strain fault rocks have been studied extensively (Bos & Spiers, 2001; Faulkner et al., 2003, Marone & Scholz, 1989).

For Opalinus Clay (OPA) deformed in laboratory, Nüesch (1991) and Jordan and Nüesch (1989)

concluded that cataclastic flow was the main deformation mechanism, with kinking and shearing on

R- and P-surfaces at the micro scale, however this was only based on observations with optical microscopy, so that grain scale processes were not resolved. Klinkerberg et al. (2009) demonstrated a correlation between compressive strength and carbonate content of two claystones; this correlation is positive for OPA but negative for Callovo Oxfordian Clays (COX). This was explained by the differences in grain size, shape, and spatial distribution of the carbonate (Klinkerberg et al. 2009), cf.

Bauer-Plaindoux et al. (1998).  Microstructural investigations using BIB-SEM and FIB-TEM in OPA

from the main fault in the Mt-Terri Underground Research Laboratory (Laurich et al., 2014, 2016)

showed that inter- and transgranular microcracking, pressure solution, clay neoformation, phyllosilicate crystal plasticity and grain boundary sliding all play an important role during the early
stages of faulting in OPA. However, simple cataclastic microstructures are rare due to the high shear
strain and there was an almost complete loss of porosity in micro- shear zones.

Digital Image Correlation (DIC) applied on images acquired during experimental deformation
provides a method to measure directly the local displacement fields (in 2D or 3D depending on the
imaging method) and quantifies locally strain over time (Lenoir et al., 2007 [Claystone, 3D, X-ray
tomography]; Bornert et al., 2010 [Claystone, 2D, optical microscopy]; Bésuelle & Hall, 2011
[Claystone, 2D, Optical microscopy]; Dautriat et al., 2011 [Carbonates, 2D, optical microscopy and
SEM]; Wang et al., 2013, 2015 [Claystone, 2D, environmental SEM]; Fauchille et al., 2015
[Claystone, 2D, Optical microscopy]; Sone et al., 2015 [Shale, 2D, SEM]). For samples with grain
sizes above micrometres, this approach allows studying processes occurring at grain scale with high
resolution (Hall et al., 2010 [Sand, 3D, X-ray tomography]; Andò et al., 2012 [Sand, 3D, X-ray
tomography]; Bourcier et al., 2012, 2013 [rock salt, 2D, optical microscopy and environmental SEM];
Wang et al., 2015 [Claystone, 2D, environmental SEM]). On claystones, DIC was used to study
swelling in environmental SEM (Wang et al., 2013, 2015) to measure strain between the clay matrix
and non-clay minerals.

Microstructural studies in naturally compacted mudrocks are currently in rapid development, enabled
by the development of ion beam milling tools (e.g. Focussed Ion Beam [FIB] and Broad Argon Ion
Beam [BIB]) which allow imaging of mineral fabrics and porosity down to nm- scale in very high
quality cross sections with SEM and TEM (Lee et al., 2003; Desbois et al., 2009, 2011, 2013, 2016;
Loucks et al., 2009; Curtis et al., 2010; Heath et al., 2011; Klaver et al., 2012; Keller et al., 2011, 2013;
Houben et al., 2013, 2014; Hemes et al., 2013, 2015; Laurich et al., 2014; Warr et al., 2014; Song et
al., 2016). Serial sectioning allows reconstruction of microstructure in 3D (Keller et al., 2011, 2013;
Milliken et al., 2013; Hemes et al., 2015), and cryogenic techniques can image the pore fluid in the
samples and avoid artefacts produced by drying (Desbois et al., 2013, 2014; Schmatz et al., 2015).

Previous work has shown that the mechanical properties of Callovo Oxfordian Clays (COX) do not
only depend on the fraction and mineralogy of the clay but also on water content and texture (Bauer-
Plaindoux et al., 1998). Chiarelli et al. (2000) showed that COX is more brittle with increasing calcite
content and more ductile with increasing clay content and proposed two deformation mechanisms:
plasticity induced by slip of clay sheets and induced anisotropic damage as indicated by microcracks
at the interface between grains and matrix, however they provided little microstructural evidence to
support this. Gasc-Barbier et al., (2004), Fabre et al., (2006), Chiarelli et al., (2003), Fouché et al.,
(2004) report that the COX has an unconfined compressive strength of 20 to 30 MPa and a Young's
modulus of 2 to 5 GPa. In the context of underground storage of radioactive wastes, these papers try to
predict the mechanical evolution of COX over the period of thousands of years. The effects studied include creep, pore-pressure dissipation, swelling, contraction, chemical effects, pressure solution and
force of crystallization. Although these papers develop elaborate constitutive laws, they provide very
limited microstructural observations. The need for micromechanical observations was already
recognized by Yang et al. (2012) and Wang et al., (2013, 2015). From Digital Image Correlation
(DIC) applied on optical and ESEM images, these authors have shown how heterogeneous strain fields
correlate with microstructure and recognized shear bands and tensile microcracks.

For highly overconsolidated claystones from the Variscan foreland thrust belt in the Ardennes and
Eifel, Holland et al. (2006) proposed an evolutionary model starting with mechanical fragmentation of
the original fabric. In this model, the initial loss of cohesion is driven by kinking, folding and micro-
fracturing processes with an increasing porosity and permeability. Abrasion during progressive
deformation increases the amount of clay gouge, and re-sealing occurs by decrease in pore size of the
clay gouge.

In summary, deformation mechanisms in mudrocks are poorly understood especially at low strain.
Although as a first approximation the plasticity of cemented and uncemented mudrocks can be
described by effective pressure- dependent constitutive models, the full description of their complex
deformation and transport properties would be much improved by better understanding of the
microscale deformation mechanisms. There is a wide range of possible mechanisms: intra- and
intergranular fracturing, cataclasis, grain boundary sliding, grain rotation and granular flow, plasticity
of phyllosilicates and the poorly known plasticity of nano-clay aggregates with the strong role of clay-
bound water, cementation, fracture sealing and solution- precipitation.

[revised manuscript text omitted]

**181   3    Methods: BIB-SEM imaging of deformed microstructures**

After the experiments of Lenoir et al. (2007) and Bésuelle et Hall (2011), deformed samples were stored at low vacuum and room temperature in a desiccator, where they dried slowly. From these deformed samples, sub-samples were selected to represent areas with different strain history based on the DIC analysis. For COX-2MPa, three BIB cross sections were prepared around the conjugate fractures in areas with different amount of diffuse strain (at the resolution of DIC), antithetic fractures (ROI-2, ROI-3 and ROI-4; Figures 2.d, 5.b, c, d and 6) and a fourth one in a region without measurable strain (ROI-1; Figures 2.d and 5.a). For COX-10MPa, two BIB-SEM analyses were done around the single shear fracture (Figures 3.d and 5.e, f).

Sub-samples were first embedded in epoxy, extracted with a low speed diamond saw in dry conditions, pre-polished dry using SiC papers (down to P4000 grade) and BIB polished in a JEOL

SM-09010 cross-section polisher (for 8 h, $1.10^{-3} – 1.10^{-4}$ Pa, 6 kV, 150 μA) to remove a 100 μm thick layer of material interpreted to be the layer of damage after polishing with SiC papers. BIB cross- sections are all prepared parallel to the $\sigma_1$ and direction and perpendicular to the shear fracture. The

[revised manuscript text omitted]

Cosenza et al., 2007; Pineda et al., 2010; Hedan et al., 2012; Renard, 2012; Wang et al., 2013, 2015;
Desbois et al., 2014).

The DIC analysis is not affected by this because the images were acquired during deformation of
preserved (wet) samples. SEM analysis is done on samples which have been deformed and unloaded,
followed by slow drying in low vacuum and further dehydration in the high vacuum of the BIB and
SEM. In COX-10MPa, this is illustrated by Figures 3.c and 3.d. Figure 3.c shows the sample at the
end of the deformation experiment, whereas Figure 3.d shows the same sample but about 10 years
later, both imaged with X-ray. The comparison of Figures 3.c and 3.d shows that cracks developed
parallel to the bedding and that the apertures of fractures developed during the deformation became
larger. These are interpreted to result from unloading and shrinkage during drying of specimens.
Though the second sample was not scanned with X-ray in dry condition, we infer that similar changes
occurred also in COX-2MPa: by analogy, there is no reason that the clay matrix in COX-2MPa
behaves differently that in COX-10MPa.

The considerations above indicate that some fractures developed during deformation but drying
damage overprinted them. Unfortunately, BIB-SEM images (performed on dried samples) do not
provide direct information to distinguish if the visible fractures and cracks developed during
deformation (and subsequently overprinted by drying) or only by drying. However, as presented in the
following paragraphs, indirect evidence suggests that the fractures in the fragments between the arrays
of antithetic fractures and the antithetic fractures of *Type I* and of *Type II* developed during
deformation.

[revised manuscript text omitted]

**5.3 Conceptual model of microstructure development in triaxially deformed COX.**

Based on BIB-SEM microstructural observations, we propose the following sequence of micro-
mechanisms Callovo-Oxfordian clay (Figure 12):

(1) & (2) Micro-fracturing

Incipient deformation occurs by intergranular microfractures propagating in the clay matrix and,
transgranular and intragranular micro fractures in non-clay minerals, both resulting in the
fragmentation of the original fabric and in agreement with the high compressive strength of this
cemented mudstone. Intergranular micro fractures are interpreted to be initiated from pores,
propagating along weak contacts at non-clay mineral / clay matrix interfaces or along (001) cleavage
planes of phyllosilicates (Chiarelli et al., 2000; Klinkenberg et al., 2009; Den Hartog & Spiers, 2014,
Jessel et al., 2009). Here note that probably the biggest unknown at present in the micro-mechanisms
of deformation in claystones is the nature of cement bonds between grains; further work in this project
is aimed at understanding this better.

(3 & 4) Cataclastic shearing with plasticity of phyllosilicates, macroscopic failure

Further deformation occurs by frictional sliding affecting the process zone at microfracture boundaries,
and in relays between fractures. Mechanisms are abrasion and bending of phyllosilicates by cataclastic
and crystal plastic mechanisms. This is accompanied by rotation of fragments and cataclastic flow.
This stage is interpreted to start at the peak stress in the stress-strain curve, accompanied by local
dilatancy. At the specimen scale, fractures link up resulting in loss of cohesion. In restraining sections
along the fractures, reworking of the clay matrix reduces porosity and eliminates large pores, changing
the pore size distributions. The specimen suffers from a major loss of cohesion accompanied by
dilatancy and stress drop after peak stress.

(5) Resealing of the damage zone by shear and pore collapse, evolution of clay gouge

Ongoing abrasion of the fragments and comminution develop a cataclastic fabric. A full understanding
of the deformation mechanisms in cataclastic clay aggregates requires more work, but the grain sliding
(Chiarelli et al., 2000) and grain rotation between low-friction clay particles together with collapsing
of porosity is inferred because: (i) slip on the (001) basal planes of clay particles is much easier than shearing related to grain breakage (cf. Haines et al., 2013 and Crider, 2015) and (ii) residual strength observed after specimen's failure argues for sliding between low frictional clay particles (Lupini et al., 1981). At sufficently high strain this stage would correspond to the residual strength result in the resealing of initial fracture porosity by filling the fractures with clay gouge. In this stage, cataclasis of non-clay particles is expected to become less important because they are embedded in reworked clay.

The conceptual model above for microstructure evolution in triaxially deformed COX is a first look based on direct grain-scale observation of microstructures. Our ongoing studies focus on the nature of the cement and at microstructures of the damage zone at fracture tips to better understand the localization mechanisms.

**6   Conclusions**

[revised manuscript text omitted]

Ishii, E., H. Sanada, H. Funaki, Y. Sugita, and H. Kurikami (2011), The relationships among brittleness, deformation behavior, and transport properties in mudstones: An example from the Horonobe Underground Research Laboratory, Japan, J. Geophys. Res., 116, B09206, doi:10.1029/2011JB008279.

Jessell, M.W., Bons, P.D., Griera, A., Evans, L. & Wilson, C.J.L. 2009. A tale of two viscosities. Journal of Structural Geology, 31: 719-736.s

Jordan P. and Nüesch R. (1989) Deformation behavior of shale interlayers in evaporite detachment horizons, Jura overthrust, Switzerland. Journal of Structural Geology, 11(7): 859-871.

Kang M-S., Watabe Y. and Tsuchida T. (2003). Effect of Drying Process on the Evaluation of Microstructure of Clays using Scanning Electron Microscope (SEM) and Mercury Intrusion Porosimetry (MIP). Proceedings of The Thirteenth (2003) International Offshore and Polar Engineering Conference Honolulu, Hawaii, USA, May 25–30, 2003

Katz O. and Reches Z. (2004). Microfracturing, damage, and failure of brittle granites. Journal of Geophysical Research, 109, B01206.

Kaufhold A., Halisch M., Zacher G., Kaufhold S. (2016). X-ray computed tomography investigation of structures in Opalinus Clay from large-scale to small-scale after mechanical testing. Solid Earth, 7: 1171-1183.

Keller L., Schuetz P., Erni R., Rossell, M.D., Lucas, F., Gasser, Ph., Holzer L. (2013). Characterization of multi-scale microstructural features in Opalinus Clay. Microporous and Mesoporous Materials, 170 : 83-94.

Keller L.M., Holzer L., Wepf R., Gasser P. (2011). 3D geometry and topology of pore pathways in Opalinus clay: Implications for mass transport. Applied Clay Science 52: 85-95.

Kim Y-S, Peacock D.C.P, Sanderson D.J. (2004). Fault damage zones. Journal of Structural Geology, 26: 503-517.

Klaver J., Desbois G., Littke R., Urai J.L. (2015). BIB-SEM characterization of pore space morphology and distribution in postmature to overmature samples from the Haynesville and Bossier Shales, Marine and Petroleum Geology, 59: 451-466.

Klaver J., Desbois G., Urai J.L. and Littke R. (2012). BIB-SEM study of porosity of immature Posidonia shale from the Hils area, Germany. International Journal of Coal Geology, 103: 12-25.

Klinkenberg M., Kaufhold S., Dohrmann R., Siegesmund S. (2009). Influence of carbonate microfabrics on the failure strength of claystones. Engineering Geology 107: 42-54.

Kohlstedt D.L., Evans B., Mackwell S.J. (1995). Strength of the lithosphere: constraints imposed by laboratory experiments. Journal of geophysical Research, 100(B9): 17587-17602.

Laurich B., Urai J.L., Desbois G., Vollmer C., Nussbaum C. (2014). Microstructural evolution of an incipient fault zone in Opalinus Clay: Insights from an optical and electron microscopic study of ion-beam polished samples from the Main Fault in the Mont Terri underground research laboratory. Journal of Structural Geology,
67: 107–128.

Laurich B., Urai J.L., Nussbaum C. (2016). Microstructures and deformation mechanisms in Opalinus Clay:
insights from scaly clay from the Main Fault in the Mont Terri Rock Laboratory (CH).Solid Earth,
doi:10.5194/se-2016-94

[revised manuscript text omitted]

Lenoir et al., 2007;).

**Figure 2:** Results of deformation test done on sample COX-2MPa. (a): deviator stress vs. axial strain response. The red star indicates the state of sample when BIB-SEM microstructural analyses are done.

(b) and (c): incremental volumetric strain fields (VSF) and maximum shear strain fields (SSF) fields for deformation increment 1-2 and 3-4 indicated in (a) after DIC. Arrows with solid lines indicate the set of two conjugated synthetic fractures whereas the arrows with dashed lines show antithetic fractures oblique to the conjugated fractures. (d): Selection of differently strained areas (ROI)

highlighted from DIC analysis for BIB-SEM microstructural analyses. Four ROI were analysed: three at conjugate synthetic fractures in areas with different amount of diffuse strain and antithetic fractures (ROI-2, ROI-3 and ROI-4) and one in a region without measurable strain (ROI-1). After Bésuelle et al.

(2011).

**Figure 3:** Results of deformation test done on sample COX-10MPa. (a): deviator stress vs. axial strain response. The red star indicates the state of sample when BIB-SEM microstructural analyses are done.

(b): incremental maximum shear strain fields for deformation increment 1-2 and 2-3 indicated in (a)

interpreted after DIC.  (c) shows the X-ray radiography of the sample taken directly at the end of the deformation test, whereas (d) shows the X-ray radiography of the same sample but taken about 10

years after the end of the deformation: drying cracks developed following the bedding and the aperture of the single shear fracture became larger. (d) indicates also two ROI were analysed both around the single synthetic shear fracture. In (c) and (d): orientation of $s_1$ and of the bedding are indicated in red.

After Lenoir et al. (2007).

[revised manuscript text omitted]

.

**Figures**

[Figure]

**Figure 1**

[Figure]

**Figure 2**

[Figure]

**Figure 3**

[Figure]

[Figure]

**Figure 4**

[Figure]

Figure 5

**Figure 5**

[Figure]

**Figure 6**

[Figure]

**Figure 7**

[Figure]

**Figure 8**

[Figure]

a. ROI-1 COX-10MPa
σ₁
b.
Host rock
Host rock
Damaged fabric
µm
a.

b. Fig. 11
Host rock
Damaged fabric
µm c. ii
ii
Damaged fabric
µm

**Figure 9**

[Figure]

**Figure 10**

[Figure]

**Figure 11**

[Figure]

**Figure 12**